# Brief early-life motor training induces behavioral changes and alters neuromuscular development in mice

Camille Quilgars[1], Eric Boué-Grabot [2], Philippe de Deurwaerdère[1], Jean-René Cazalets[1], Florence E. Perrin[3,4], Sandrine S. Bertrand [1]*

**1** Université de Bordeaux, CNRS, INCIA, UMR 5287, Bordeaux, France, **2** Université de Bordeaux, CNRS, Institut des Maladies Neurodégénératives, UMR 5293, Bordeaux, France, **3** MMDN, Univ. Montpellier, EPHE, INSERM, Montpellier, France, **4** Institut Universitaire de France (IUF), Paris, France

\* sandrine.bertrand@u-bordeaux.fr

## Abstract

In this study, we aimed to determine the impact of an increase in motor activity during the highly plastic period of development of the motor spinal cord and hindlimb muscles in newborn mice. A swim training regimen, consisting of two sessions per day for two days, was conducted in 1 and 2-day-old (P1, P2) pups. P3-trained pups showed a faster acquisition of a four-limb swimming pattern, accompanied by dysregulated gene expression in the lateral motor column, alterations in the intrinsic membrane properties of motoneurons (MNs) and synaptic plasticity, as well as increased axonal myelination in motor regions of the spinal cord. Network-level changes were also observed, as synaptic events in MNs and spinal noradrenaline and serotonin contents were modified by training. At the muscular level, slight changes in neuromuscular junction morphology and myosin subtype expression in hindlimb muscles were observed in trained animals. Furthermore, the temporal sequence of acquiring the adult-like swimming pattern and postural development in trained pups showed differences persisting until almost the second postnatal week. A very short motor training performed just after birth is thus able to induce functional adaptation in the developing neuromuscular system that could persist several days. This highlights the vulnerability of the neuromuscular apparatus during development and the need to evaluate carefully the impact of any given sensorimotor procedure when considering its application to improve motor development or in rehabilitation strategies.

## Introduction

In altricial mammals such as rodents and humans, the spinal networks responsible for the locomotor pattern and rhythm, known as central pattern generators (CPGs), are already functional at birth, but locomotor activity cannot be expressed until several days or months due to an immature neuromuscular system [1–3]. The maturation of

**Data availability statement:** Data are available from the Research Data Gouv Institutional Data Access. https://doi.org/10.57745/WE0VK8

**Funding:** This work benefited from the support of the Laser Microdissection capture facility funded by Inserm, LabEX BRAIN ANR-10-LABX-43 and FRM DGE20061007758, thanks to M. Maitre and H. Doat of the NeuroCentre Magendie Inserm U1215. This study received financial support from the French government in the framework of the University of Bordeaux's IdEx "Investments for the Future" program / GPR BRAIN_2030 (CQ, SSB, EBG, JRC, PDD). The funders had no role in study design, data collection and analysis, decision to publish, or preparation of the manuscript

**Competing interests:** The authors have declared that no competing interests exist.

the neuromuscular apparatus and the acquisition of adult motor patterns have been proposed to rely on the 'neuronal group selection' theory, which reconciles the nature and nurture theories. Genetically predetermined neural networks are built by spontaneous activity during fetal development and refined after birth through sensory-driven motor experiences that select and stabilize the most efficient and favorable neuronal networks [4,5]. Changes or alteration in motor experiences during development should therefore have major impact on neuromuscular circuits. Animals deprived of normal patterns of neuromuscular activation through various experimental protocols, such as immobilization, limb disuse, sciatic nerve crush, or pharmacological blockage of nervous activity during development, have been shown to exhibit alterations in the developing spinal cord and muscles [6–12] that could have long-lasting deleterious impacts [13,14]. In contrast, the impact of increased motor activity and related functional consequences on motor networks during the developmental period is largely ignored.

In adults, the beneficial role of exercise in both physiological and pathophysiological conditions is now widely accepted and well documented. Intense/endurance exercise training performed in adult rodents triggers physiological adaptations in spinal motoneurons (MNs) associated with changes in neuromuscular transmission efficacy and cortical ultrastructure [15–20]. Activity-based rehabilitation strategies are currently used after spinal cord injury (SCI). Numerous studies have addressed the plastic processes involved in exercise-induced functional recovery in the injured spinal cord in both humans and animals. see for examples: [21–28].

A general consensus seems to emerge that during development, physical activity also leads to a general improvement in both motor and cognitive functions. The impact of physical exercise in children appears however highly dependent on the nature and intensity of the motor activity performed [29–32].

In rodents, the spinal cord networks and muscles undergo major reconfiguration during the first two postnatal weeks [3,33–36]. What could be the effects of increased activity in motor structures during this highly plastic and critical period of development? Would it trigger an acceleration of maturation or lead to maladaptive plasticity? In the present study, we addressed these questions by submitting newborn mice to motor training consisting of short swimming sessions during the first two postnatal days. Using a combination of techniques, we demonstrated that a two-day motor training performed shortly after birth was sufficient to trigger behavioral, physiological, structural, and genomic changes in the neuromuscular system. Our data reveal the molecular and cellular mechanisms that sustain part of experience-driven plasticity during the highly sensitive period of spinal cord and muscle development.

## Results

### Motor training accelerated the acquisition of the four-limb motor pattern for swimming

Rodents exhibit spontaneous swimming abilities at birth. In the first two postnatal weeks, the swimming activity pattern in rodents follows a specific temporal

sequence, providing a developmental index [37–39]. Between P0-P1, pups predominantly use their forelimbs for swimming, transitioning to the use of all four limbs between P3-P14 and acquire the adult pattern, characterized by the exclusive use of hindlimbs around P15. Consequently, we chose a swimming motor training approach to evaluate how heightened motor activity influences motor development (Fig 1A, see Material and Methods). We compared the swimming pattern in trained pups over motor training sessions with age-matched untrained mice. In untrained animals, the initial swimming session (S1, green boxes and bars in Fig 1A and 1B1) served as a habituation phase to water and was marked by short swimming sequences alternating with floating periods. A second session (S2, red boxes and bars in Fig 1A and 1B2) was conducted to elicit longer swimming episodes. A motor score was determined based on the type and maximal number of legs used predominantly during the first 7 seconds of each session, with the highest value (4) indicating the use of all four limbs (Table 1). Only well-defined swimming patterns were taken into account for analysis, and animals performing mixed patterns during the 7 seconds of motor scoring were ignored. The number of animals included in the analysis for each training session could therefore differ. The Fig 1B1 shows that during S1, some P1 pups failed to swim (motor score of 0) in both animal group tested (~ 25% untrained, ~ 32% trained). However, the majority of them were already able to swim with their forelimbs and even with 1 or 2 of their hindlimbs (Fig 1B1). During S1, we observed a gradual increase in the use of three or four limbs with age in both untrained and trained mice. On the second day of training during S1, some trained pups showed no swimming activity, whereas all untrained P2 pups were able to swim (Figs 1B1, S1, trial 3 versus training 3: $c2 = 18.5$, $df = 3$; Chi-square, $p < 0.01$; trial 4 versus training 4: $c2 = 9.4$ $df = 3$; Chi-square, $p = 0.025$). During S2 (Fig 1B2), a significant difference in the distribution of motor scores was observed between untrained and trained pups across all tested time points (S2, trial 1 versus training 1: $c2 = 33.6$, $df = 3$; Chi-square, $p < 0.0001$; trial 2 versus training 2: $c2 = 32.7$ $df = 3$; Chi-square, $p < 0.0001$; trial 3 versus training 3: $c2 = 20.8$, $df = 3$; Chi-square, $p < 0.0001$; trial 4 versus training 4: $c2 = 30.2$ $df = 3$; Chi-square, $p < 0.0001$). During S2, all trained P2 mouse pups were capable of swimming during the second day, whereas around 20% of untrained P2 mice failed to swim (Fig 1B2, motor score of 0). The proportions of pups using three or four limbs to swim during S2 increased with age in both groups but were consistently higher in trained animals compared to untrained ones. Notably, the four-limb pattern represented the most frequent swimming pattern used by P2 mouse pups during S2 of training 4 (Fig 1B2). Fig 1C depicts the percentage of trained mouse pups using four limbs for swimming (motor score of 4) in the five different sessions across the four training periods. This percentage reached a plateau during S3 and then slightly decreased during the S4 and S5 likely due to animal fatigue. The proportion of animals using the four-limb pattern significantly increased between Training 1 and Trainings 3 and 4 (Fig 1C, Friedman test, $p = 0.001$).

The body weight of both untrained and trained pups significantly increased with age but appeared not significantly different between the two animal groups (P1: trained $1.4 \pm 0.01$ g, $n = 181$; untrained $1.3 \pm 0.01$ g, $n = 243$; P2: trained $1.6 \pm 0.08$ g, $n = 175$; untrained $1.5 \pm 0.01$, $n = 256$ and P3: trained $1.8 \pm 0.02$, $n = 171$; untrained $1.8 \pm 0.02$, $n = 256$; two-way ANOVA followed by uncorrected Fisher's LSD post-tests, interaction: $F_{2,1,310} = 6$, $p = 0.0026$, age: $F_{2,1,310} = 437$, $p < 0.0001$, training: $F_{1,1,310} = 2$, $p = 0.16$).

To investigate potential changes in postural control, we compared the righting reflex of P3 trained ($n = 39$) and untrained ($n = 36$) pups. As shown in Fig 1D, the time to right was comparable between the two groups (untrained: $11.3 \pm 1.3$ s, trained: $11.9 \pm 2$ s; Mann-Whitney, $p = 0.22$). The number of animals that succeeded in righting was also not significantly different (untrained: 69.8%, trained: 77.9%, $p = 0.15$; Chi-square). However, we observed that trained pups attempted to right for longer periods compared to untrained ones (Fig 1D; untrained: $19 \pm 2.4$ s, trained: $33.3 \pm 3.7$ s until abandon, Mann-Whitney, $p < 0.01$).

Taken together, these results suggest that a short motor training performed twice a day in P1-P2 animals facilitated the acquisition of the four-limb motor pattern for swimming (S1 Video) and appears to increase resistance to fatigue during the righting test in P3 mouse pups.

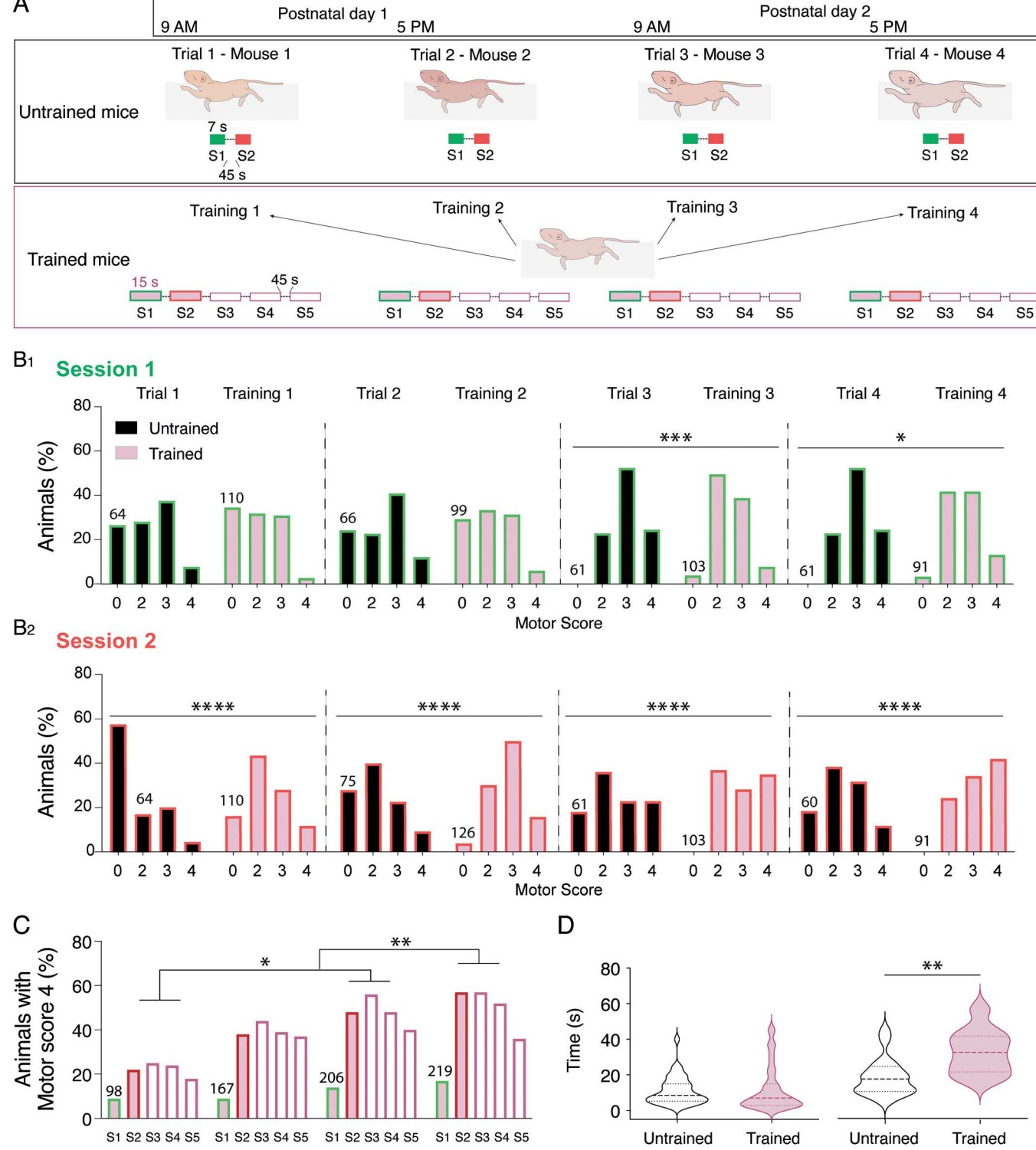

**Fig 1. Swim motor training accelerates the acquisition of the four-limb motor pattern in P3 trained mice. A.** Schematic representation of the experimental protocol. For each training (1 to 4), trained mice realized five successive swimming sessions (S1 to S5) of 15 seconds separated by 45

second breaks twice a day (9 AM and 5 pm) on post-natal day 1 (P1) and P2. Untrained mice were tested only once either at P1 9 AM or P1 5 pm or P2 9 AM or P2 5 pm during two swimming sessions of 7 seconds each (S1 green box and S2 red box). The greatest number of limbs used during the first 7 seconds of each session was scored (motor score, Table 1 P1-P3) **B.** Histograms of the percentage of untrained (trial, black filles bars) and trained (training, purple filled bars) animals according to the motor score assessed during the first (bars with green border; B1) and second swimming session (bars with red border; B2). The number of animals included in the analysis appears on histogram bars. **C.** Percentage of trained mice with a motor score of 4 (4 four limbs used to swim) during the different swim trainings. The total number of animals with a motor score of 4 appears on histogram bars. **D.** Violin plots of the time to right (left panel) during the righting test and time until abandon (right panel) in untrained (black, n = 36 pups) and trained (purple, n = 39 pups) mice. *p < 0.05, **p < 0.01, ***p < 0.001, ****p < 0.0001 two-way ANOVA analysis, followed by uncorrected Fisher's LSD post-tests in B and C and Mann-Whitney tests in **D.** Underlying data can be found in the S1 Data Sheet.

**Table 1. Score assessment scale for scoring swim ability in mice.**

| Score | Motor pattern used in majority during the session |
|---|---|
| P1-P3 During training | |
| 0 | No swimming |
| 2 | Two forelimbs only use to swim |
| 3 | Two forelimbs and one hindlimb |
| 4 | Both forelimbs and hindlimbs |
| P5-P12 | |
| 4 | Both forelimbs and hindlimbs use to swim |
| 3 | One forelimb used to swim, the other one in extension |
| 2 | The two forelimbs are alternately used for swimming or placed in extension. |
| 1 | Two forelimbs in extension |

## Motor training induces changes in gene expression in the lateral motor column

We then explored whether early motor training influences gene expression in MNs. RNA-seq data were compared between the lateral motor column (LMC) of the lumbar spinal cord of five untrained and five trained P3 HB9-GFP mice. We analyzed 15387 genes. We observed significant modifications in the expression of 478 genes, setting a threshold of 2-fold change (FC), in trained pups compared to untrained ones, with 292 genes upregulated and 186 downregulated (Fig 2A, p-value < 0.05). S1 Table presents all differentially expressed genes (DEGs) identified in this analysis. Many DEGs encode transcription factors (Fig 2B) and homeobox genes (Fig 2C), including *Notch4* and *Reg3b*, which are involved in MN differentiation and myelinization [40–42], as well as *FoxP2* and *Neurod6*, which are crucial for spinal inhibitory neuron development [43]. Additionally, several DEGs are associated with apoptotic processes (Fig 2C). A significant number of transcripts encoding ion channels, particularly $K^+$ channels, were found to be dysregulated (Fig 2D), along with genes involved in synaptic function and neurotransmission (Fig 2D). Network-based enrichment analysis (S1A Fig) revealed that upregulated genes are involved in multiple intracellular signaling pathways crucial for neural development. Conversely, downregulated genes are associated with processes related to muscle contraction and neuromuscular junctions (NMJ) development. Furthermore, downregulated genes are associated with neurogenesis, synaptogenesis, and potassium transport (S1A Fig). Go processes analysis showed that many upregulated genes are linked to developmental processes, anatomical structure and system development (S1B Fig). This suggests that a short-time and early motor training performed during a high plastic developmental period of the central nervous system modifies gene expression pattern in the LMC and may influence the development and function of MNs and related systems such as adaptations in neuromuscular junctions and skeletal muscles.

## Motor training impacts the intrinsic membrane properties of lumbar MNs

We explored whether swim training influenced the physiology of lumbar MNs by examining first their intrinsic membrane properties. To simplify, MNs recorded from trained and untrained animals will hereafter be referred to as "trained MNs" and

PLOS Biology

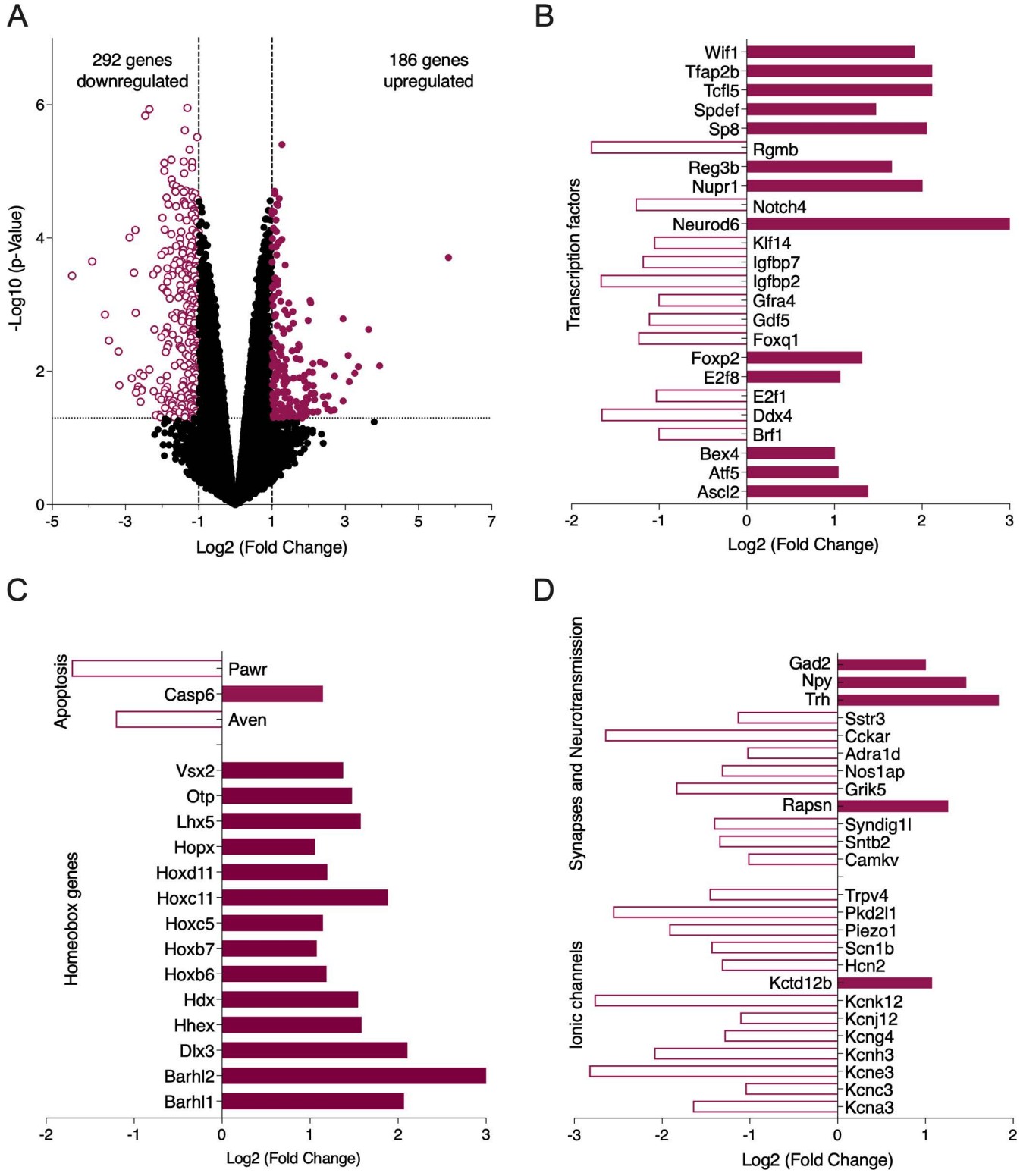

**Fig 2. Differential gene expression in the LMC of P3 untrained and trained mice. A.** Volcano plot showing RNA-seq data of downregulated (purple open circles) and upregulated (purple filled circles) genes, using a 2-fold change (FC) and a p-value (< 0.05) as selection criteria, indicated by dashed lines. Bar plots showing significant fold changes (log2) in the expression profiles of transcripts encoding transcription factors **(B)**, homeobox and

apoptosis-related proteins **(C)**, as well as ion channels, synaptic, and neurotransmission-related proteins **(D)**. *Kcna3*: potassium voltage-gated channel, shaker-related subfamily, member 3. *Kcnc3*: potassium voltage gated channel, Shaw-related subfamily, member 3. *Kcne3*: potassium voltage-gated channel, Isk-related subfamily, gene 3. *Kcnh3*: potassium voltage-gated channel, subfamily H (eag-related), member 3. *Kcng4*: potassium voltage-gated channel, subfamily G, member 4. *Kcnj12*: potassium inwardly-rectifying channel, subfamily J, member 12. *Kcnk12*: potassium channel, subfamily K, member 12. *Kctd12b*: potassium channel tetramerization domain containing 12b. *Hcn2*: hyperpolarization-activated, cyclic nucleotide-gated K$^+$ 2. *Scn1b*: sodium channel, voltage-gated, type I, beta. *Piezo1*: piezo-type mechanosensitive ion channel component 1. *Pkd2l1*: polycystic kidney disease 2-like 1. *Trpv4*: transient receptor potential cation channel, subfamily V, member 4. *Camkv*: CaM kinase-like vesicle-associated. *Sntb2*: syntrophin, basic 2. *Syndig1l*: synapse differentiation inducing 1 like. *Rapsn*: receptor-associated protein of the synapse. *Grik5*: glutamate receptor, ionotropic, kainate 5 (gamma 2). *Nos1ap*: nitric oxide synthase 1 (neuronal) adaptor protein. *Adra1d*: adrenergic receptor, alpha 1d. *Cckar*: cholecystokinin A receptor. *Sstr3*: somatostatin receptor 3. *Trh*: thyrotropin releasing hormone. *Npy*: neuropeptide Y. *Gad2*: glutamic acid decarboxylase 2. *Ascl2*: achaete-scute family bHLH transcription factor 2. *Atf5*: activating transcription factor 5. *Bex4*: brain expressed X-linked 4. *Brf1*: BRF1, RNA polymerase III transcription initiation factor 90 kDa subunit. *Ddx4*: DEAD box helicase 4. *E2f1*: E2F transcription factor 1. *E2f8*: E2F transcription factor 8. *Foxp2*: forkhead box P2. *Foxq1*: forkhead box Q1. *Gdf5*: growth differentiation factor 5. *Gfra4*: glial cell line derived neurotrophic factor family receptor alpha 4. *Igfbp2*: insulin-like growth factor binding protein 2. *Igfbp7*: insulin-like growth factor binding protein 7. *Klf14*: Kruppel-like transcription factor 14. *Neurod6*: neurogenic differentiation 6. *Notch4*: notch 4. *Nupr1*: nuclear protein transcription regulator 1. *Reg3b*: regenerating islet-derived 3 beta. *Rgmb*: repulsive guidance molecule family member B. *Sp8*: trans-acting transcription factor 8. *Spdef*: SAM pointed domain containing ets transcription factor. *Tcfl5*: transcription factor-like 5 (basic helix-loop-helix). *Tfap2b*: transcription factor AP-2 beta. *Wif1*: Wnt inhibitory factor 1. *Barhl1*: BarH like homeobox 1. *Barhl2*: BarH like homeobox 2. *Dlx3*: distal-less homeobox 3. *Hhex*: hematopoietically expressed homeobox. *Hdx*: highly divergent homeobox. *Hoxb6*: homeobox B6. *Hoxb7*: homeobox B7. *Hoxc5*: homeobox C5. *Hoxc11*: homeobox C11. *Hoxd11*: homeobox D11. *Hopx*: HOP homeobox. *Lhx5*: LIM homeobox protein 5. *Otp*: orthopedia homeobox. *Vsx2*: visual system homeobox 2. *Aven*: apoptosis, caspase activation inhibitor. *Casp6*: caspase 6. *Pawr*: PRKC, apoptosis, WT1, regulator. Raw data are available under the following link: https://doi.org/10.57745/WE0VK8.

"untrained MNs," respectively. The input membrane resistance (Rin), rheobase, spike threshold, and various action potential (AP) parameters of MNs showed no significant differences between trained and untrained animals (Table 2). Mature MNs frequently exhibit a small calcium-dependent after-depolarization (ADP) following the action potential (AP) (Fig 3A1) [36,44]. Although the ADP amplitude did not show significant differences between trained and untrained MNs (Table 2), a higher proportion of MNs displayed an ADP after training compared to control conditions (Fig 3A2; c2 = 5.9 df = 2; Chi-square, p = 0.015). The AHP amplitude and duration (both half-width and half-decay time) was significantly increased in trained MNs compared to untrained MNs (Fig 3A3–3A6), while the AHP rise time was not altered (Fig 3A5).

To determine whether training and after-hyperpolarization (AHP)-related changes affect the instantaneous frequency-current ($f$-I) relationships of MNs, we applied a series of depolarizing current steps in cells held at −60 mV (Fig 3B1). Some tested MNs emitted only one or two spikes instead of sustaining firing activity during a series of depolarizing currents. These neurons were observed in a slightly but significantly increased proportion in trained MNs compared to

**Table 2. Training impact on electrical properties of P3 lumbar MNs.** Rin: input resistance. Rheobase: lowest intensity of current injected in MNs to elicit an action potential (AP). AP threshold: voltage measured at the foot of the AP. AP amplitude: measured between the resting membrane potential and the AP peak. AP half width: time spent by the potential > 50% of the AP maximum amplitude. AP rise time: time spent by the potential between 10% and 90% of the AP maximum amplitude. AP half decay time: time spent by the potential between the AP maximum amplitude and the 50% decreasing amplitude. ADP: after-depolarization potential. All values are means ± SEM. The number of MNs recorded is indicated between brackets. ns: no significantly different. P-values are obtained from Mann-Whitney test or T-test statistical analysis, depending of the normal distribution of data sets. Underlying data can be found in the S1 Data Sheet.

| | Untrained | Trained | p-value |
|---|---|---|---|
| RIN (MΩ) | 72.6 ± 4.6 (63) | 59.4 ± 3.6 (49) | 0.06 (ns) |
| RHEOBASE (PA) | 411.1 ± 38.8 (34) | 466.2 ± 51.2 (24) | 0.37 (ns) |
| AP THRESHOLD (MV) | 36.58 ± 1.71 (34) | 33. 95 ± 1.65 (24) | 0.27 (ns) |
| AP AMPLITUDE (MV) | 88.35 ± 2.34 (34) | 87.87 ± 3.58 (24) | 0.91 (ns) |
| AP HALF WIDTH (MS) | 2.84 ± 0.18 (34) | 3.38 ± 0.51 (24) | 0.99 (ns) |
| AP RISE TIME (MS) | 81.44 ± 11.02 (34) | 56.66 ± 9.61 (24) | 0.11 (ns) |
| AP HALF DECAY TIME (MS) | 1.33 ± 0.08 (34) | 1.48 ± 0.16 (24) | 0.39 (ns) |
| ADP AMPLITUDE (MV) | 15.46 ± 0.91 (45) | 13.66 ± 0.82 (46) | 0.14 (ns) |

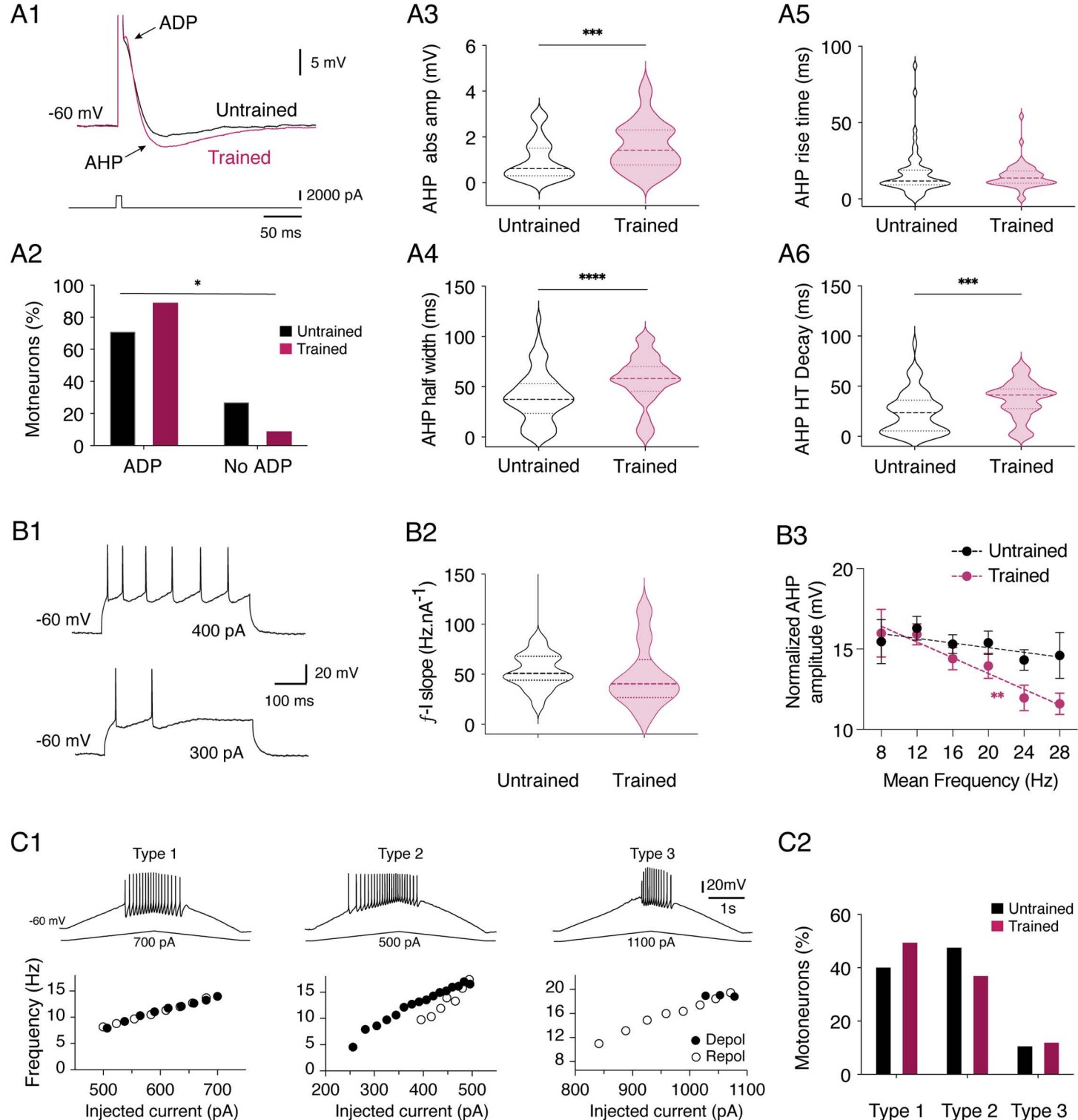

**Fig 3. Training impact on intrinsic membrane properties of lumbar MNs. A.** Representative traces of the after depolarization (ADP) and after hyperpolarization (AHP) expressed after an action potential (A1 upper trace, truncated onto the trace) induced by a short current pulse (bottom trace) in an untrained (black trace) MN and a trained MN (purple trace) held at −60 mV in current clamp condition. Percentage of untrained (black) and trained

(purple) MNs expressing ADP or not. *p < 0.05, Chi square test (A2). Violin plots of the absolute values of AHP amplitude (AHP abs ampl) measured in untrained (black) and trained (purple) MNs (A3). Same data representation as in A3 for the AHP half width (A4), the AHP rise time (A5) and the AHP half decay time (AHP HT Decay; A6). N = 58 untrained MNs and n = 52 trained MNs; * significantly different; ***p < 0.001, and ****p < 0.0001, Mann-Whitney tests. **B.** Representative traces of the firing activity induced in a MN during the application of a 300 and a 400 pA depolarizing current pulse injection (B1). Violin plot of the *f*/I slope computed in the different MNs tested in untrained (black, n = 20 MNs) and trained (purple, n = 10 MNs) animals (B2). Plot of the normalized AHP amplitude as a function of the mean frequency of spike activity obtained during series of depolarizing current pulses in untrained (black dots) and trained (purple dots) animals. The dashed lines correspond to the linear fitting. * Significantly different; **p < 0.01, Pearson test (B3). **C.** Representative traces of the three different types of discharge recorded from MNs during triangular ramp current injection (upper panels) and corresponding plots (bottom panels) of the instantaneous firing frequencies as a function of the current injected during the depolarization (filled circles) and the repolarization (open circles) phase (C1). Percentage of MNs expressing the Type 1, 2 or 3 profile of discharge in untrained (black bars, n = 27) and trained (purple bars, n = 16) MNs (C2). Underlying data can be found in the S1 Data Sheet.

untrained ones (untrained: n = 14 MNs out of 42 spiking MNs, trained: n = 21 MNs out of 36 spiking MNs; c2 = 4.9, df = 1; Chi-square test, p = 0.03). In MNs exhibiting a sustained firing rate, the slope of the *f*-I relationship was computed. The average slope of the *f*-I relationship did not show any significant change in MNs of trained pups compared to untrained ones (Fig 3B2 and Table 3). When expressing the mean AHP amplitude as a function of the firing frequency of MNs during current pulse application, we observed no correlation in untrained animals, whereas a significant negative correlation was observed in MNs of trained mice (Fig 3B3).

Based on the categorization established previously [45–47], MNs of types 1, 2, and 3 were identified during triangular current pulse application (Fig 3C1). The proportion of each subtype was similar in untrained and trained pups (Fig 3C2, c2 = 2.7 df = 2; Chi-square, p = 0.25). Some MNs, unable to fire during the ramp current, were found in the same proportions in untrained and trained mice (untrained n = 21 MNs out of 48 MNs, trained n = 19 MNs out of 35 MNs; c2 = 9 df = 1; Chi-square, p = 0.34). Take, together, these data indicate that the overall excitability of MNs is unchanged in trained mice, but that training impact both AHP parameters and involvement in the regulation of the firing rate of P3 MNs.

RNAseq data indicate a downregulation of several genes associated with channel activity in trained mice (Fig 2D). Specifically, we observed changes in genes coding for subunits of inwardly rectifying K$^+$ (Kir) channels and hyperpolarization-activated cyclic nucleotide-modulated cation non-selective (HCN) channels (Fig 2D). Functional Kir channels consist of Kir 2.1, 2.2, 2.3, and 2.4 subunits in both homomeric and heteromeric combinations see for review[48]. We observed a downregulation in the expression of the gene coding for the Kir 2.2 subunit in trained mice compared to untrained animals (Fig 2D, *Kcnj12* log2(FC) = −1.11, FDR = 0.02). In contrast, no significant differences were found for Kir 2.1 (log2(FC) = 0.34, FDR = 0.61), Kir 2.4 (log2(FC) = −0.71, FDR = 0.15), and no expression of Kir 2.3 was detected in our data. HCN channels constitute a family of non-selective cation channels structurally similar to K$^+$ channels, supporting the hyperpolarization-activated current I$_H$ [48]. HCN2 gene

**Table 3. Training impact of the lumbar MN discharge properties.** ISI: initial interspike interval corresponding of the instantaneous discharge between the first two AP. ISI slope: ISI frequency-current relationship. ESFF: early-state firing frequency representing the mean 3 first ISI. ESFF slope: ESFF frequency-current relationships. Current min: minimum current injected in MNs used for slope analysis of spike-frequency adaptation (SFA) parameters. Current max: maximum current injected in MNs used for slope analysis of SFA parameters. All values are means±SEM. The number of MNs recorded is indicated between brackets. ns: no significantly different. P-values are obtained from Mann-Whitney test or *T* test statistical analysis, depending of the normal distribution of data sets. Underlying data can be found in the S1 Data Sheet.

| | Untrained | Trained | p-value |
|---|---|---|---|
| ISI SLOPE (HZ/NA) | 65.31 ± 6.99 (20) | 62.18 ± 11.78 (11) | 0.82 (ns) |
| ESFF SLOPE (HZ/NA) | 58.21 ± 6.70 (20) | 42.86 ± 9.99 (11) | 0.07 (ns) |
| SSFF SLOPE (HZ/NA) | 51.00 ± 4.66 (20) | 40.56 ± 5.40 (11) | 0.16 (ns) |
| CURRENT MIN (NA) | 0.44 ± 0.04 (20) | 0.39 ± 0.09 (11) | 0.26 (ns) |
| CURRENT MAX (NA) | 0.64 ± 0.04 (20) | 0.63 ± 0.08 (11) | 0.55 (ns) |

expression was significantly downregulated (log2(FC) = −1.32; FDR = 0.04) following training (Fig 2D). We investigated whether these transcriptomic changes result in functional alterations in the Kir and $I_H$ currents expressed in MNs. To activate Kir current, long voltage ramps from −40 mV to −150 mV were applied in MNs (Fig 4A1). The chord conductance of induced currents was assessed at membrane potentials equidistant from the $K^+$ equilibrium potential (Erev): at Erev + 40 mV for the outward current and Erev −40 mV for the inward current (Fig 4A1). The measured Erev value was not significantly different between untrained and trained MNs (untrained: −107 ± 0.8 mV, n = 40 versus trained: −108 ± 0.85 mV, n = 36; Mann-Whitney test, p = 0.55). Consistent with the rectifying characteristics of the Kir current, the inward component of the current was significantly larger than the outward component in both untrained and trained MNs (Fig 4A2; untrained: 12 ± 1 pS Erev + 40 mV versus 17 ± 1.3 pS, Erev −40 mV, n = 40; trained: 15 ± 1.2 pS Erev + 40 mV versus 22 ± 1.9 pS Erev −40 mV, n = 36; two-way ANOVA followed by uncorrected Fisher's LSD post-tests, interaction: F1,74 = 3.6, p = 0.06, voltage: F1,74 = 118, p < 0.0001). The outward component of the Kir current showed a tendency to increase in trained MNs, but was not significantly different (Fig 4A2). In contrast, the inward chord conductance calculated in trained MNs was significantly higher than in untrained MNs (Fig 4A2; two-way ANOVA followed by uncorrected Fisher's LSD post-tests, interaction: F1,74 = 3.6,

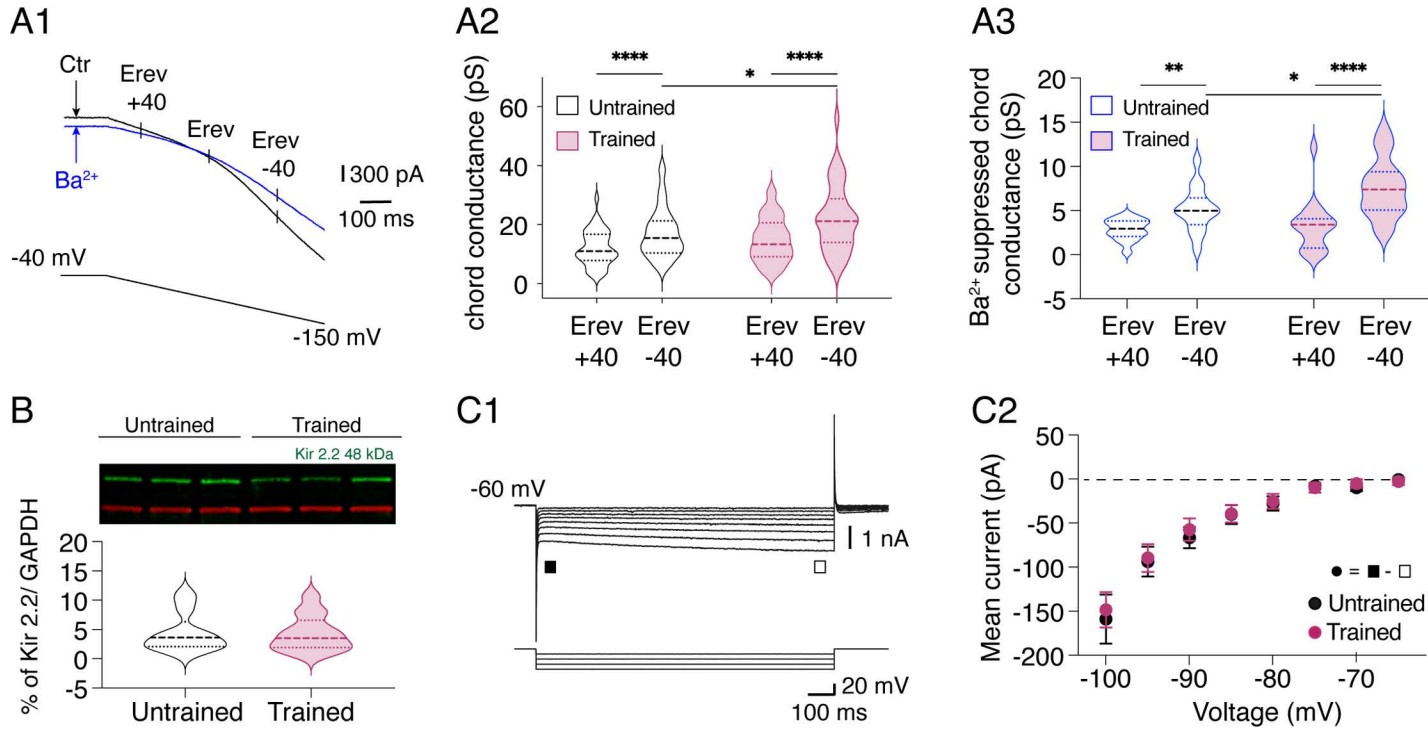

**Fig 4. Training impact on the Kir and $I_H$ currents in P3 lumbar MNs. A.** Representative traces of the current recorded during the application of a long voltage ramp from −40 mV to −150 mV in MNs held at −40 mV in control condition (black trace) and in the presence of 500 µm $Ba^{2+}$ (blue trace) (A1). The mean cord conductance of the current was measured at membrane potentials equally distant from the reversal potential (Erev), Erev+40 and Erev-40. Violin plots of the chord conductances measured in untrained (black, n = 40) and trained (purple, n = 36) MNs (A2). Violin plots of the chord conductance of the current suppressed by $Ba^{2+}$ in untrained (black, n = 15) and trained (purple, n = 17) MNs (A3). * Significantly different; *p < 0.05, **p < 0.01, ***p < 0.001, two-way ANOVA analysis, followed by uncorrected Fisher's LSD post-tests. **B.** Representative Kir2.2 (green) and GAPDH (red) bands from ventral lumbar spinal cords of untrained and trained P3 mice (top panel), with the quantitative results of western blotting analysis (bottom panel). **C.** Sample membrane current traces obtained in response to negative voltage steps in a MN held at −60 mV in voltage clamp conditions. The $I_H$ current was computed by subtracting the instantaneous current (■) from the steady state (□) (E1). Mean instantaneous I-V curves derived from the current responses generated by a series of voltage steps in untrained (black dots, n = 14) and trained (purple dots, n = 17) MNs (E2). Underlying data can be found in the S1 Data Sheet.

p = 0.06, training: F1,74 = 4.5, p = 0.03). As the voltage ramp used could activate both Kir and $I_H$ currents, we used barium (Ba²⁺, 500 μM) to selectively block the Kir current (Fig 4A1, blue trace) and compared the Ba²⁺-suppressed currents between untrained and trained MNs. As shown in Fig 4A3, the current blocked by Ba²⁺ displayed a significantly greater inward component than outward component in both trained and untrained MNs. The outward chord conductance of the inhibited current was similar between untrained (2.8 ± 0.3 pS, n = 15) and trained MNs (3.43 ± 0.7 pS, n = 17). However, the inward chord conductance was significantly larger in trained MNs compared to untrained MNs (untrained: 5.2 ± 0.65 pS, trained: 7.5 ± 0.8 pS; two-way ANOVA followed by uncorrected Fisher's LSD post-tests, interaction: F1,30 = 5, p = 0.03, voltage: F1,30 = 57, p < 0.0001, training: F1,30 = 2.8, p = 0.1). We examined whether these functional changes were coupled to changes in the expression of the Kir 2.2 subunit protein in the ventral lumbar part of the spinal cord by western blotting (Fig 4B, n = 13 untrained and trained mice). Protein level of Kir 2.2 was unchanged between untrained and trained mice (Fig 4B; untrained: 4.4 ± 1% of Kir 2.2/ GAPDH versus trained: 4.3 ± 1% of Kir 2.2/ GAPDH, Mann-Whitney test, p = 0.92). These data suggest that the observed alterations in the expression of genes coding for Kir 2.2 subunits and the functional changes in the Kir current in trained MNs are not associated with modifications in the overall protein expression of Kir 2.2 in the ventral spinal cord.

To examine the impact of the swim training on the $I_H$ current, a series of hyperpolarizing voltage pulses were applied. The instantaneous current evoked was measured immediately after the capacitive transient (filled square in Fig 4C1) and the steady-state current at the end of the hyperpolarizing pulse (open square in Fig 4C1). The difference between the steady state and the instantaneous current has been shown to corresponded to the $I_H$ current [49,50]. The amplitude of $I_H$ computed in untrained and trained MNs was not significantly different (Fig 4C2, n = 46 untrained MNs, n = 32 trained MNs; two-way ANOVA followed by uncorrected Fisher's LSD post-tests, interaction: F7,532 = 0.44, p = 0.9, voltage: F7,532 = 111, p < 0.0001, training: F1,76 = 0.7, p = 0.4) suggesting that the observed transcriptomic changes in HCN2 subunit following training do not impact the physiology of the $I_H$ current in lumbar MNs.

## Modification of the activity-dependent plasticity expressed at VLF-MN synapses by the motor training

We have previously shown the existence of developmentally regulated, activity-dependent synaptic plasticity (ADSP) at synapses between glutamatergic axons conveyed in the ventrolateral funiculus (VLF) of the spinal cord and MNs following high-frequency stimulations of the VLF [34]. The expression of ADSP is highly sensitive to changes in neuronal network activity [51,52]. We investigated whether early motor training could impact ADSP expression at VLF-MN synapses. When paired-pulse stimulations (50 ms interval) were applied to the VLF, excitatory postsynaptic currents (VLF-EPSCs) were triggered in lumbar MNs and paired-pulse facilitation (PPF) of VLF-EPSCs was observed (Fig 5A), as previously described [34]. The PPF ratio value (VLF-EPSC2/VLF-EPSC1) did not show significant changes due to training (untrained MNs: 1.3 ± 0.04, n = 25; trained MNs: 1.4 ± 0.03, n = 37; Mann-Whitney, p = 0.73), indicating no major alteration in the release probability of VLF-MN synapses following training. High-frequency stimulation (HFS, 50 Hz, 2 s) applied to VLF axons (VLF-HFS) resulted in short-term depression (STD) or long-term depression (LTD) of VLF-EPSC amplitudes at VLF- MN synapses in untrained P3 mice (Fig 5B), consistent with previous findings [34]. Under our experimental conditions, the majority of tested MNs exhibited LTD (n = 19 out of 25; Fig 5B and 5D). Interestingly, the same stimulating protocol induced both LTD (Fig 5C, n = 19) and STD (Fig 5C, n = 11) in trained mice but also failed to modulate VLF-EPSC amplitudes at some VLF- MN synapses (Fig 5C, No Plasticity, n = 7), a phenomenon previously described as specific to older (P8-P12) MNs under control conditions [34]. The proportion of MNs expressing LTD, STD, or No Plasticity differed significantly between untrained and trained mice (Fig 5D, c2 = 6.39, df = 2; Chi-square, p = 0.04). This result suggests that training induced changes in the ADSP expressed at VLF-MN synapses, which are likely part of the functional adaptations triggered by this slight but significant increase in motor activity.

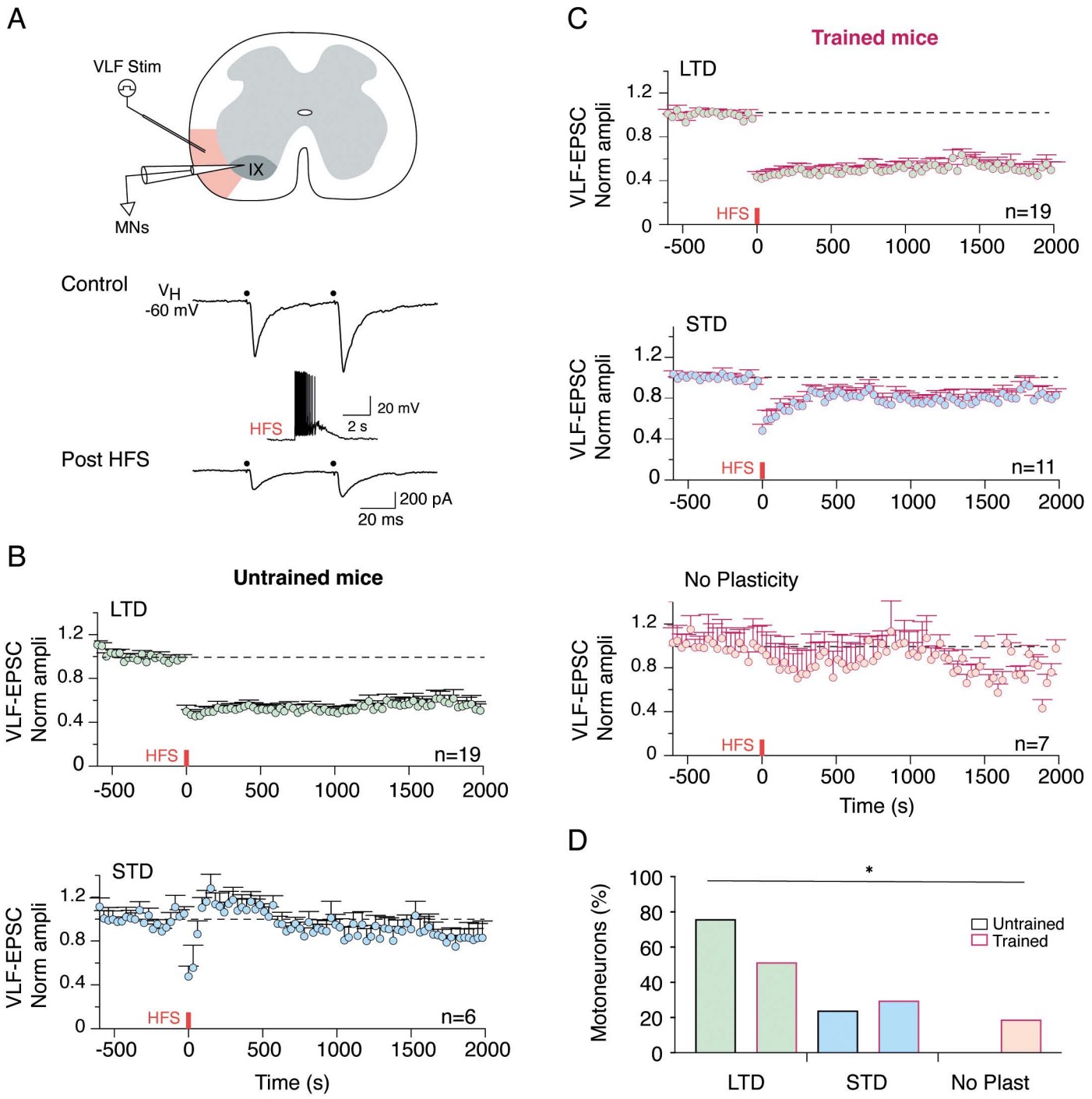

**Fig 5. Training impact on the ADSP expression at VLF-MNs synapses. A.** Schematic diagram of the experimental protocol and sample traces of EPSCs elicited during paired-pulse stimulations of VLF axons (VLF stim, 50 ms interval, black dots) in lumbar MNs held at −60mV before and after VLF-HFS (50 Hz, 2 s, middle trace). **B.** Pooled data average time courses of normalized VLF-EPSC amplitudes in VLF-MNs synapses expressing LTD (green dots) or LTD (blue dots) in untrained mice. **C.** Pooled data average time courses of normalized VLF-EPSC amplitudes in VLF-MNs synapses expressing LTD (green dots), STD (blue dots) or no plasticity (red dots) after VLF-HFS in trained mice. **D.** Percentage of different ADSPs expressed by untrained (bars with black border, n=25) and trained (bars with purple border, n=37) MNs. * Significantly different. *p<0.05, Chi square test. Underlying data can be found in the S1 Data Sheet.

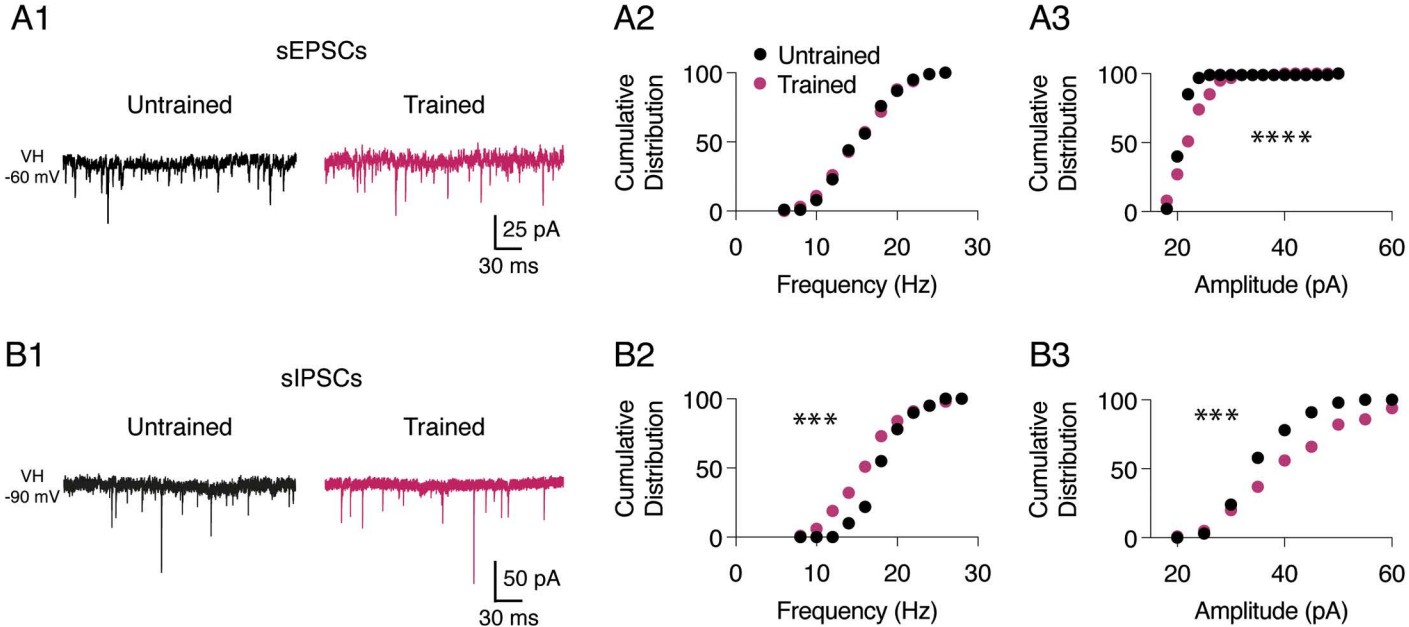

**Fig 6. Training impact on spontaneous network activity in spinal motor networks of P3 mice. A.** Representative traces of spontaneous excitatory post-synaptic currents (sEPSCs) recorded in the presence of strychnine and gabazine in an untrained (black trace) and a trained (purple trace) MN held at −60 mV ($V_H$ −60 mV) (A1). Cumulative distributions of sEPSC frequency (A2) and amplitude (A3) recorded in untrained (black dots, n = 20) and trained (purple dots, n = 14) MNs. **B.** Representative traces of spontaneous inhibitory post-synaptic currents (sIPSCs) recorded in the presence of DNQX and AP5 in an untrained (black trace) and a trained (purple trace) MN held at −90 mV ($V_H$ −90 mV). Cumulative distributions of sIPSC frequency (B2) and amplitude (B3) recorded in untrained (black dots, n = 24) and trained (purple dots, n = 11) MNs. * Significantly different. ***p < 0.001, and ****p < 0.0001, Kolmogorov–Smirnov test. Underlying data can be found in the S1 Data Sheet.

## Motor training impacts the spinal network excitability and monoaminergic content

We then investigated whether motor training has impacted the general excitability of the spinal network by recording and comparing spontaneous excitatory and inhibitory post-synaptic currents (Fig 6A1 and 6B1; sEPSCs and sIPSCs, respectively) from untrained and trained MNs. Cumulative distribution analysis revealed no significant changes in the frequency of sEPSC events (KS test, p = 0.9; Fig 6A2) but a significant decrease in the amplitude of sEPSCs in trained MNs (n = 14) compared to untrained MNs (n = 20; KS test, p < 0.0001; Fig 6A3). Moreover, we observed more frequent but smaller sIPSP events in trained MNs (n = 11) compared to the untrained MN cohort (n = 24; KS test, p = 0.0002 and 0.0008, respectively). These changes in the pattern of spontaneous activity indicates that the spinal network activity is modified after motor training in P3 MNs.

The monoaminergic descending fibers, crucial for the maturation of the locomotor network [53], start to invade the lumbar enlargement before birth and gradually increase in density until the end of the second postnatal week [54]. Using HPLC analysis (chromatogram in Fig 7A), we investigated whether motor training modified the spinal content of monoamines, specifically 5-HT, dopamine (DA), and noradrenaline (NA), in trained P3 mice but also in P10 animals to evaluate potential long-lasting effects. The DA content in the entire lumbar spinal cord tissues remained unchanged regardless of age or experimental condition tested (Fig 7B; two-way ANOVA followed by uncorrected Fisher's LSD post-tests, interaction: $F_{1,66} = 4.5$, p = 0.04, training: $F_{1,66} = 0.05$, p = 0.8, age: $F_{1,66} = 0.008$, p = 0.9). While the 5HT spinal content did not show significant age-related changes in untrained animals, we observed a notable increase between P3 and P10 in trained mice. Training tended to elevate the 5-HT spinal content at P3 and significantly increased it at P10

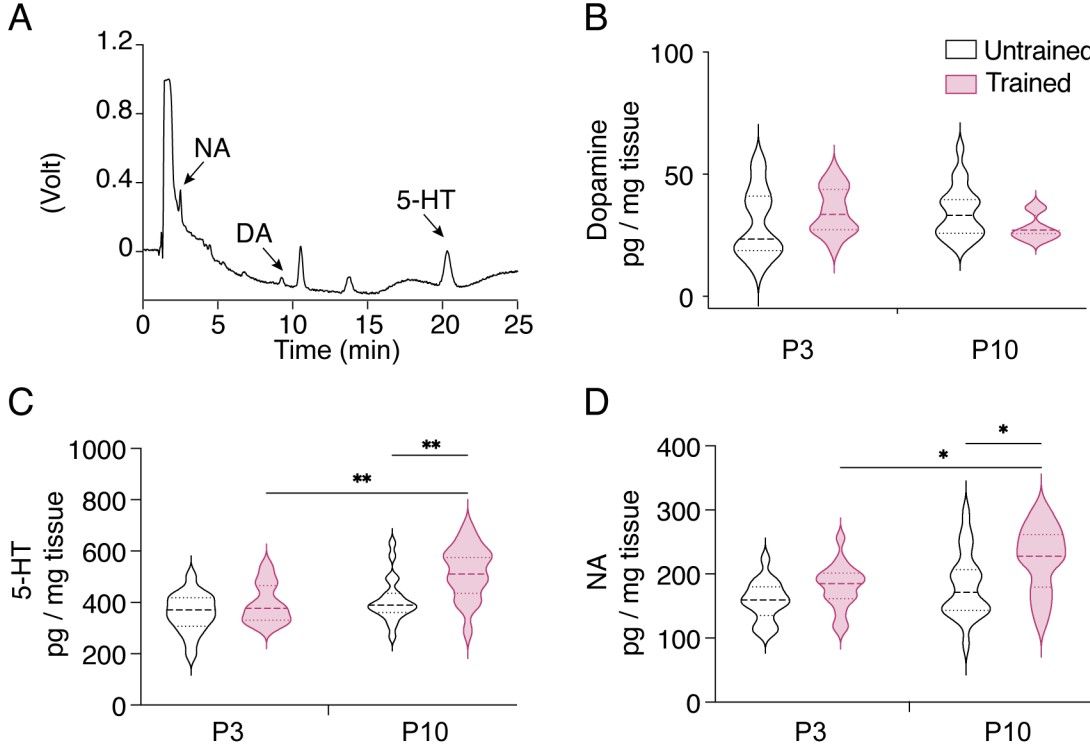

**Fig 7. HPLC analysis of noradrenaline (NA), dopamine (DA) and serotonin (5-HT) contents in the ventral lumbar spinal cord of P3 (n = 16 and 17) and P10 (n = 20 and 17) untrained and trained mice, respectively. A.** Representative chromatogram of the monoamine oxidation over time. **B.** Violin plots of dopamine content in the ventral lumbar spinal cords from P3 and P10 untrained (black) and trained (purple) mice. **C.** Same as in B for 5-HT content analysis. **D.** Same as in B for NA content analysis. *p < 0.05, **p < 0.01, two-way ANOVA analysis, followed by uncorrected Fisher's LSD post-tests. Underlying data can be found in the S1 Data Sheet.

when compared to untrained mice (Fig 7C; two-way ANOVA followed by uncorrected Fisher's LSD post-tests, interaction: $F_{1,68} = 2.5$, $p = 0.1$, training: $F_{1,68} = 15.8$, $p = 0.0002$, age: $F_{1,68} = 10.4$, $p = 0.002$). Similar results were found for NA (Fig 7D, two-way ANOVA followed by uncorrected Fisher's LSD post-tests, interaction: $F_{1,67} = 0.8$, $p = 0.4$, age: $F_{1,67} = 9$, $p = 004$, training: $F_{1,67} = 10$, $p = 0.002$). These data suggest that motor training has long lasting impacts on the spinal contents of 5-HT and NA.

## Acceleration of myelinization specifically in spinal motor areas following motor training

In the next series of experiments, we aimed to determine whether the early training impacts the myelination of axons in the spinal cord of newborn mice. The process of axonal myelination that occurs during the first two postnatal weeks in rodents, is highly responsive to experience and can be either stimulated or altered by the manipulation of axonal activity [55,56]. Fluoromyelin staining revealed the presence of myelin sheaths in the white matter of the dorsal and ventral commissures, as well as in the ventrolateral area in the spinal cord of both untrained and trained P3 mice (Fig 8A1). Analysis of fluoromyelin-positive structures (Fig 8A2 and see Material and Methods) demonstrated a significant increase in myelin sheath thickness in the ventrolateral area of trained mice compared to untrained animals ($p = 0.007$, Mann-Whitney test), with no notable difference observed in either commissure (Fig 8B1, $p = 0.08$ and 0.5, Mann-Whitney tests, for the dorsal and ventral commissure, respectively). Additionally, comparison of axons revealed that the area of myelinated axons was unchanged in the dorsal commissure, but significantly larger in both the ventral commissure and ventrolateral area in trained animals compared to untrained ones (Fig 8B2, statistical results of the two-way ANOVAs in Table 4). The graph

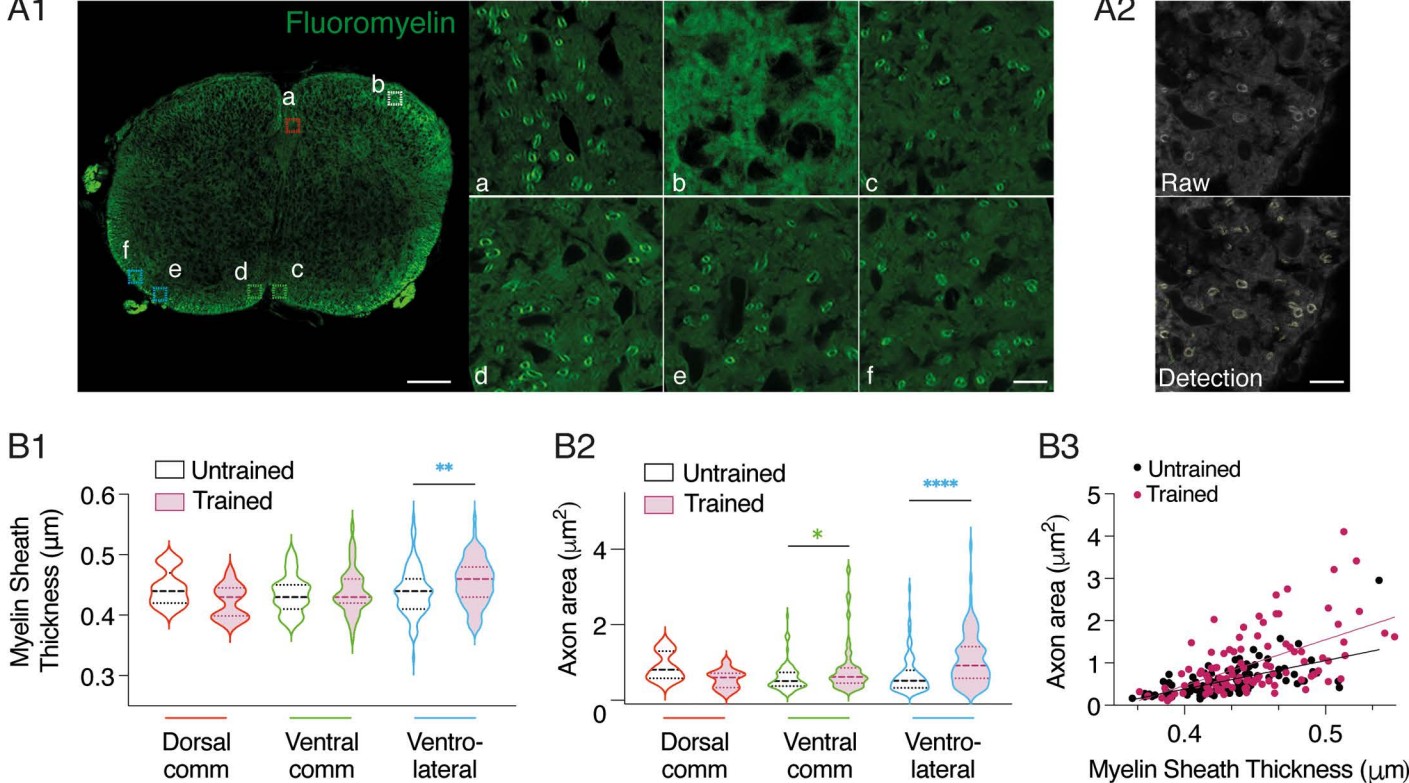

**Fig 8. Impact of short motor training on axonal myelination in the lumbar spinal cord. A.** Representative fluorescence microscopy image of a transverse spinal cord section depicting fluoromyelin labeling (green channel). Magnified views of the white matter in the dorsal commissure (a, red hatched box), dorsal horn (b, white hatched box), ventral commissure (c and d, green hatched boxes), and ventral horn (e and f, blue hatched boxes) (A1). Representative image of raw data and after the detection of fluoromyelin labeling using Fiji (A2). Calibration bars: 20 µm and 5 µm in magnifications **B.** Violin plots illustrating the thickness of the myelin sheath (B1) and axon area (B2) in the different regions analyzed for trained (purple filled violins) and untrained (unfilled violins) mice. *p<0.05, **p<0.01, ****p<0.0001, two-way ANOVA analysis, followed by uncorrected Fisher's LSD post-tests. Graph plot of the myelin sheath thickness as a function of the axon area in trained (purple dots) and untrained (black dots) mice (B3). The lines represent the linear fitting. Four trained and 4 untrained mice were used for this analyze. Underlying data can be found in the S1 Data Sheet.

plot in Fig 8B3 shows that regardless of the experimental condition, the sheath of myelinated axons in P3 mice exhibit a strong positive correlation with the axon area (p<0.001, Spearman test for both trained and untrained condition). These data suggest that motor training influences axonal growth and myelination specifically in the ventro-lateral part of the lumbar spinal cord.

## Morphological changes induced by motor training at the muscular level in P3 pups

The RNAseq analysis, showed that some dysregulated genes of P3 trained mice were implicated in processes linked to muscle contraction and development of NMJs. Consequently, we evaluated the potential impact of early motor training at the muscular level by comparing the NMJs and muscle fibers subtypes between trained and untrained mouse pups. To assess the pre- and postsynaptic morphology of NMJs, neurofilament labeling was used for MN axons, and nicotinic cholinergic receptors (nAChRs) were labeled. Additionally, laminin was used to delineate muscle fiber membranes (Fig 9A1). In the early developmental stages, NMJs appear as simple plaque-like structures (Fig 9A2), distinct from the complex pretzel-like structures observed in adults. As extensor hindlimb muscles have been demonstrated to display a delayed developmental pattern compared to flexor muscles [57], we examined NMJs in the *tibialis anterior*, an ankle flexor muscle

**Table 4. Statistical table of two-way ANOVA analysis for axonal myelination and area, neuromuscular junction (Acetylcholine plaque) parameters, muscle parameters and motor activity quantification considering anatomical position or age and training as main factors. Number of P3 animals used: Myelin analysis: 4 trained and 4 untrained mice; Ach Plaques: 5 trained and 4 untrained mice; Muscles: 4 trained and 5 untrained and Motor activity: 14 trained and 11 untrained mice.**

| | Interaction | | | Spinal cord region | | | Training | | |
|---|---|---|---|---|---|---|---|---|---|
| | F | Df | p | F | Df | p | F | Df | p |
| **Myelin Sheath Thickness** | 3.2 | 2, 358 | 0.04 | 3.8 | 2, 358 | 0.02 | 0.04 | 1, 358 | 0.8 |
| **Axon area** | 5 | 2, 358 | 0.007 | 3.8 | 2, 358 | 0.02 | 1.5 | 1, 358 | 0.2 |
| | **Interaction** | | | **Muscle type** | | | **Training** | | |
| | F | Df | p | F | Df | p | F | Df | p |
| **Ach Plaque** | | | | | | | | | |
| Volume | 4.2 | 1, 276 | 0.04 | 5.6 | 1, 276 | 0.02 | 0.05 | 1, 276 | 0.8 |
| Area | 4.1 | 1, 276 | 0.04 | 14.1 | 1, 276 | 0.0002 | 1.5 | 1, 276 | 0.2 |
| Area/Volume | 1.4 | 1, 276 | 0.2 | 0.0002 | 1, 276 | 1 | 2.3 | 1, 276 | 0.1 |
| Compactness coefficient | 1.7 | 1, 276 | 0.2 | 25 | 1, 276 | <0.0001 | 2.2 | 1, 276 | 0.1 |
| Sphericity coefficient | 0.2 | 1, 276 | 0.7 | 13.3 | 1, 276 | 0.0003 | 10.2 | 1, 276 | 0.002 |
| NMJ contact surface | 0.1 | 1, 116 | 0.7 | 2.7 | 1, 116 | 0.1 | 0.03 | 1, 116 | 0.8 |
| **Muscles** | | | | | | | | | |
| Area | 1.1 | 2, 86 | 0.3 | 25 | 2, 86 | <0.0001 | 20 | 1, 86 | <0.0001 |
| Fiber area | 2.4 | 2, 86 | 0.09 | 118 | 2, 86 | <0.0001 | 7.3 | 1, 86 | 0.008 |
| Density of fibers | 9.7 | 2, 86 | 0.0002 | 359 | 2, 86 | <0.0001 | 3.7 | 1, 86 | 0.06 |
| MyHC Embryo | 0.9 | 2, 30 | 0.4 | 4.4 | 2, 30 | 0.02 | 0.02 | 1, 30 | 0.9 |
| MyHC I | 0.9 | 2, 25 | 0.4 | 5.2 | 2, 25 | 0.01 | 0.6 | 1, 25 | 0.4 |
| MyHC IIB | 10.2 | 2, 29 | 0.0004 | 42.7 | 2, 29 | <0.0001 | 11.2 | 1, 29 | 0.002 |
| | **Interaction** | | | **Age** | | | **Training** | | |
| | F | Df | p | F | Df | p | F | Df | p |
| **Motor Activity** | 0.9 | 3, 92 | 0.4 | 10.7 | 3, 92 | <0.0001 | 0.1 | 1, 92 | 0.7 |

and in a group of posterior muscles, including the g*astrocnemius lateralis, gastrocnemius medialis, soleus*, and *plantaris*, known primarily as extensor muscles. The two-way ANOVA analysis, followed by uncorrected Fisher's LSD post-tests (Table 4), revealed that, with the exception of the area/volume ratio (Fig 9B3), the volume (Fig 9B1), area (Fig 9B2) as well as compactness (Fig 9B3) and sphericity (Fig 9B4) coefficients of AchR plaques are significantly different between *tibialis* and posterior muscles in untrained P3 mice. In trained mice, only the compactness and sphericity coefficients of AchR plaques were different between the muscles tested (Fig 9B5). Furthermore, training significantly reduced the area of AchR plaques in the *tibialis* (Fig 9B2) and the area/volume ratio in posterior muscles (Fig 1B3). A significant increase in the compactness coefficient in posterior muscles (Fig 9B4) and in the sphericity coefficient of AchR plaques in both muscles (Fig 9B5) was also observed in trained animals compared to untrained ones. No significant changes were observed in the surface contact surface between the AChR plaque and the MN axons labeled with neurofilament between untrained and trained mice (Fig 9C, results of the two-way ANOVA analysis, followed by uncorrected Fisher's LSD post-tests in the Table 4). Overall, these data indicate that motor training leads to modified organization and morphology of the AchR plaques in both flexor and extensor muscle groups without changing the surface contact between pre- and post-synaptic partners at the NMJ.

Skeletal muscles are remarkably adaptable structures that modify their fiber profile by adjusting the composition and expression levels of myosin heavy chain (MyHC) isoforms in response to changes in muscle physiology or function. During development, a precise pattern of MyHC protein expression has been documented, with specific MyHC subtypes exclusively expressed during developmental stages [35]. We investigated the impact of training on the overall

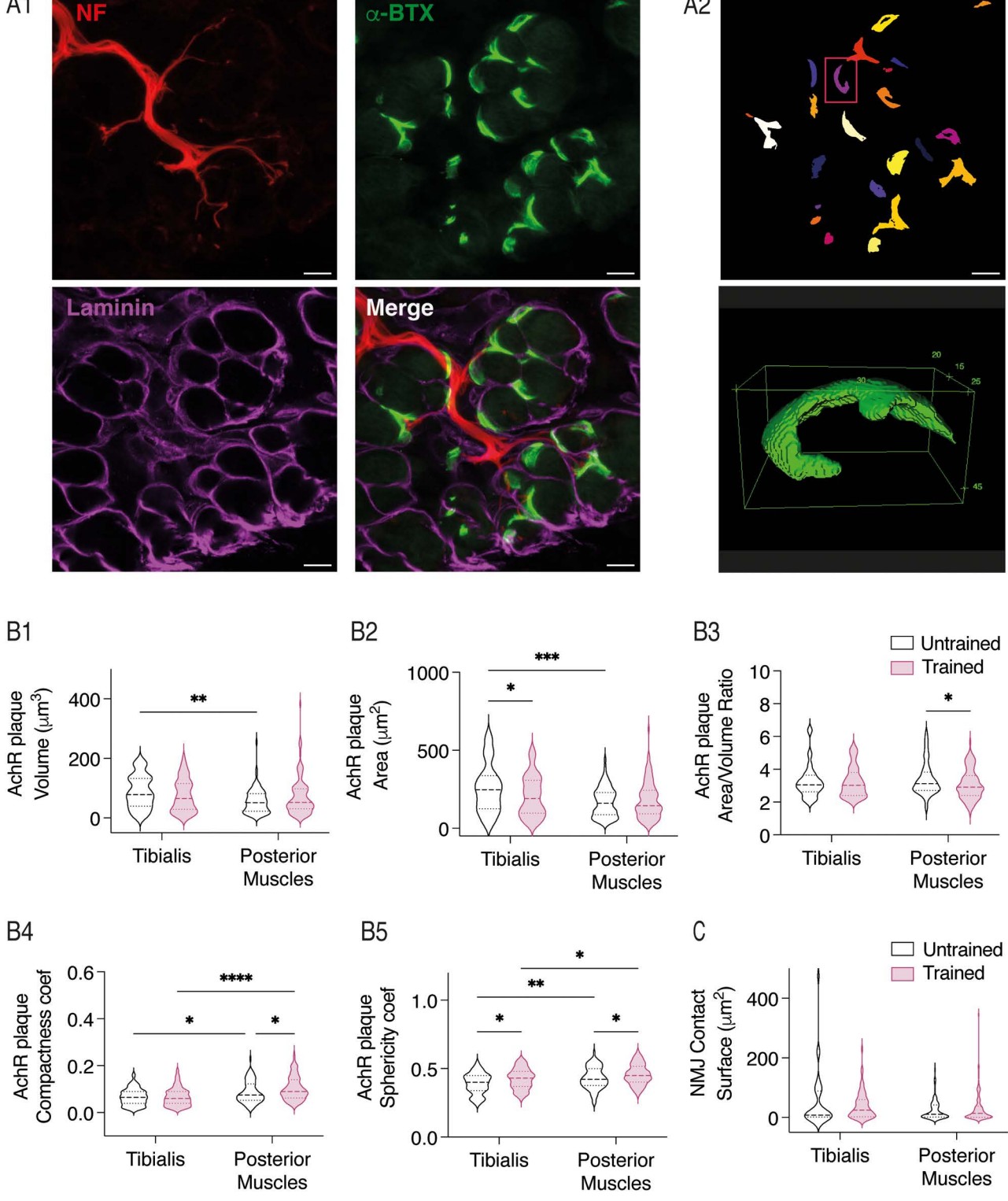

**Fig 9. Neuromuscular junctions after training in P3 hindlimb muscles. A.** Representative confocal images of transverse *tibialis anterior* section from a P3 mouse, showing neurofilament (NF), α-bungatoxin (α-BTX,) and laminin labeling (A1). Representative image of NMJ detection (upper panel) and 3D reconstruction (lower panel) with Fiji (A2). Calibration bars: 10 μm. **B.** Violin plots present the volume (B1), area (B2), area/volume ratio (B3),

compactness coefficient (B4) and sphericity coefficient (B5) of AchR clusters in the *tibialis anterior* and posterior muscles for trained (purple) and untrained (black) mice. **C.** Violin plots of the surface contact between motoneuron terminals and NMJs in trained (purple) and untrained (black) mice. Analyzed muscles were from 5 trained mice and 4 untrained mice. *p < 0.05, **p < 0.01, ***p < 0.001, ****p < 0.0001 two-way ANOVA analysis, followed by uncorrected Fisher's LSD post-tests. Underlying data can be found in the S1 Data Sheet.

morphological parameters of hindlimb muscles, along with the expression of four different MyHC isoforms in P3 mice. We considered three easily identifiable types of muscles on coronal sections of pup hindlimbs: the *tibialis anterior* (TA), the *gastrocnemius lateralis* (GL), and the *gastrocnemius medialis* (GM; see Fig 10A1). The results of the uncorrected Fisher's LSD post-tests from the two-way ANOVA analysis used for data comparison are presented in the Table 4. In both untrained and trained animals, the GL exhibited a larger area, greater muscle fiber area, and higher fiber density compared to the other two muscles. Interestingly, training induced a small but significant decrease in the area of both the overall muscle and fibers in the TA and GL (Fig 10B1–10B2), as well as a significant increase in fiber density in the TA and a decrease in the GL (Fig 10B3). As illustrated in Fig 10A2 and as demonstrated by others [58], the predominant MyHC found in muscles at this developmental stage is the embryonic MyHC. In our experiments, attempts to label the neonatal form using a previously published antibody directed against the neonatal MyHC form proved unsuccessful. MyHC I was detected in less than 10% of the fibers in the three muscles analyzed. The presence of the MyHC IIB isoform was observed exclusively in the TA muscle (Fig 10A2 and 10C3), while the MyHC IIA was entirely absent in P3 hindlimb muscles (Fig 10A2). Two-way ANOVA analysis, followed by uncorrected Fisher's LSD post-tests (Table 4), indicated that the impact of training on MyHC composition in hindlimb muscles of P3 mice is limited, with only a notable decrease in the percentage of MyHC IIB observed in the TA (Fig 10C3).

### Early motor training impacts the development of postural control during the two first postnatal weeks

As mentioned earlier, the swimming activity pattern in rodent pups follows a specific temporal sequence characterized by a gradual switch from exclusive use of forelimbs until P3, a transitional period involving the use of all four limbs, to exclusive use of hindlimbs around P15 [37–39]. All the pups tested previously were sacrificed at P3. Two untrained (n = 11 pups) and two trained (n = 14 pups) littermates were dedicated to the analysis of motor development during the two first postnatal weeks. Almost all animals tested exhibited a motor score of 4 (using all four limbs, Table 1 P5-P12) at P5 and P7 (Fig 11A1) when placed in water. Between P10 and P12, some pups swam only with their hindlimbs, leaving their fore-limbs extended, while others extended one forelimb while using the other for swimming. By P14, all tested animals used only their hindlimbs to swim, with their forelimbs extended (motor score of 1, Table 1 P5-P12 and Fig 11A1). We observed no significant difference in the evolution of the swimming pattern between untrained and trained mouse pups during the developmental period tested. However, all the trained mice were capable of raising their heads above water during swimming at P10, whereas only approximately 40% of untrained pups could do so (Fig 11A2; Chi-square test, p < 0.0001). When the motor activity of pups aged P5 to P12 was tested on a solid surface, both the untrained and trained groups showed a similar progression in terms of head elevation capability (Fig 11B1), but displayed significantly different distri-butions in acquiring postural control of the shoulders and pelvis (Fig 11B2–11B3). At P5, a higher proportion of trained mice was capable of raising their shoulders compared to untrained animals; however, these proportions were reversed at P7 and became similar by P10 (Fig 11B2, Chi-square test, p < 0.0001). While similar proportions were observed between trained and untrained pups at P5, all trained pups were able to raise their pelvis at P7, whereas only 73% of the untrained pups exhibited this capability (Fig 11B3, Chi-square test, p = 0.002). To further investigate the development of postural control, we compared the righting reflex of trained (n = 14) and untrained (n = 11) pups during the first two postnatal weeks. At P7, trained mice righted significantly faster than untrained mice (Fig 11B4, Mann-Whitney test, p = 0.001). By P10, both populations immediately righted themselves after being placed on their backs. The overall motor activity, measured over a

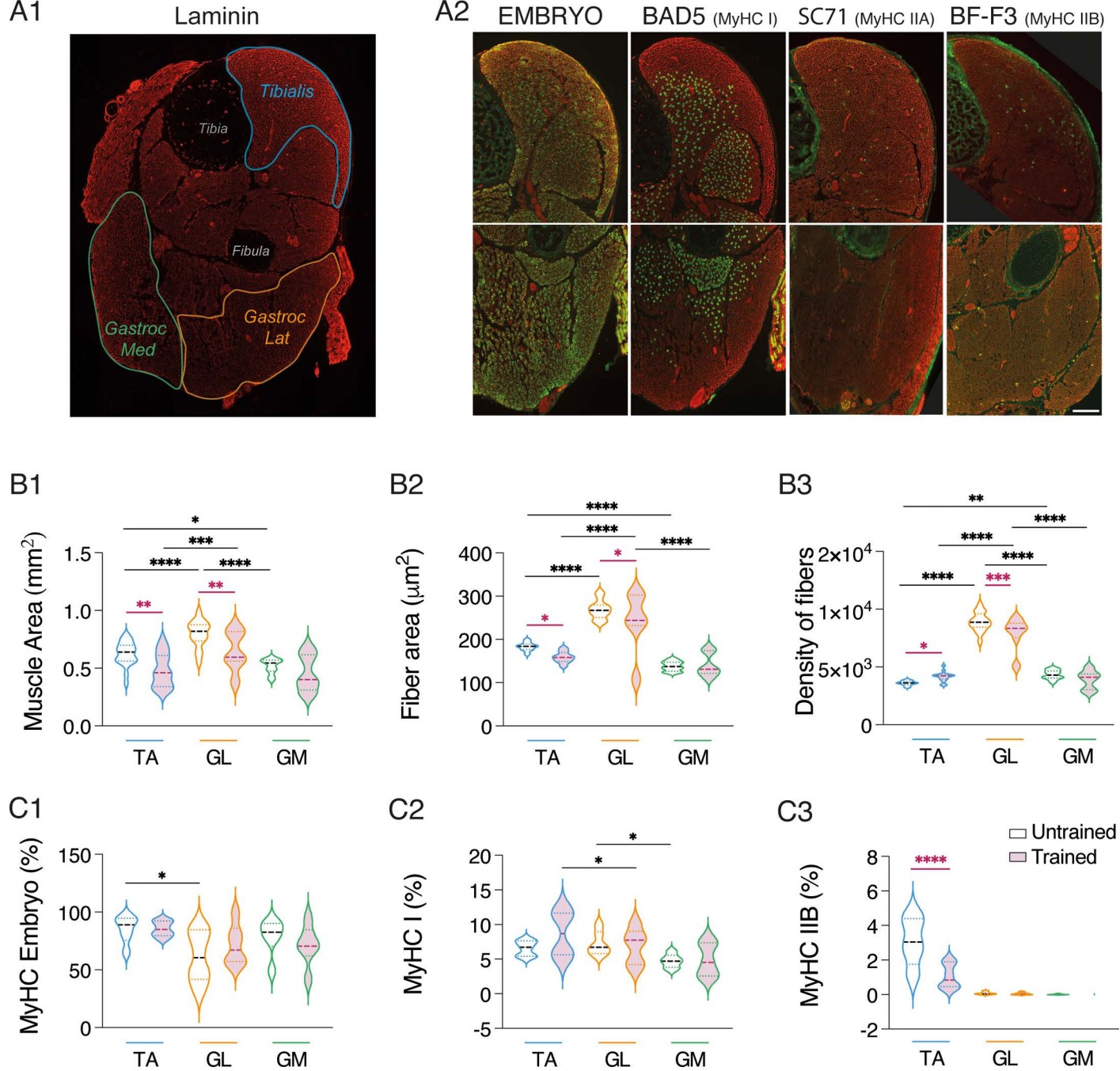

**Fig 10. Impact of training on myosin heavy chain subtypes in P3 hindlimb muscles A.** Cross-section through the hindlimb of a P3 mouse labeled by immunofluorescence for laminin (A1). The *tibialis anterior* is outlined in blue, the lateral *gastrocnemius* (Gastroc Lat) in orange, and the medial *Gastrocnemius* (Gastroc Med) in green. Representative images of the immunofluorescence labeling obtained in the *tibialis* (upper panel) and lateral *gastrocnemius* muscles (lower panel) for Embryo Myosin heavy chain (MyHC), MyHC type I BAD5 antibody, MyHC IIA SC71 antibody, and MyHC IIB BF-F3 antibody (A2). Calibration bar: 200 μm. **B.** Violin plots of the muscle area (B1), fiber area (B2), and fiber density (B3) in the three muscles analyzed in untrained (unfilled violins) and trained (purple-filled violins) P3 mice. **C.** Violin plots of the percentage of MyHC Embryo (C1), MyHC I (C2), and MyHC IIB in the three muscles analyzed in untrained (unfilled violins) and trained (purple-filled violins) P3 mice. *Tibialis* and *gastrocnemius* muscles were isolated from 4 trained mice and 5 untrained mice. *p < 0.05, **p < 0.01, ***p < 0.001, ****p < 0.0001, two-way ANOVA analysis, followed by uncorrected Fisher's LSD post-tests. Purple asterisks correspond to significant effects associated with training, and black asterisks to significant effects linked to muscle subtype. Underlying data can be found in the S1 Data Sheet.

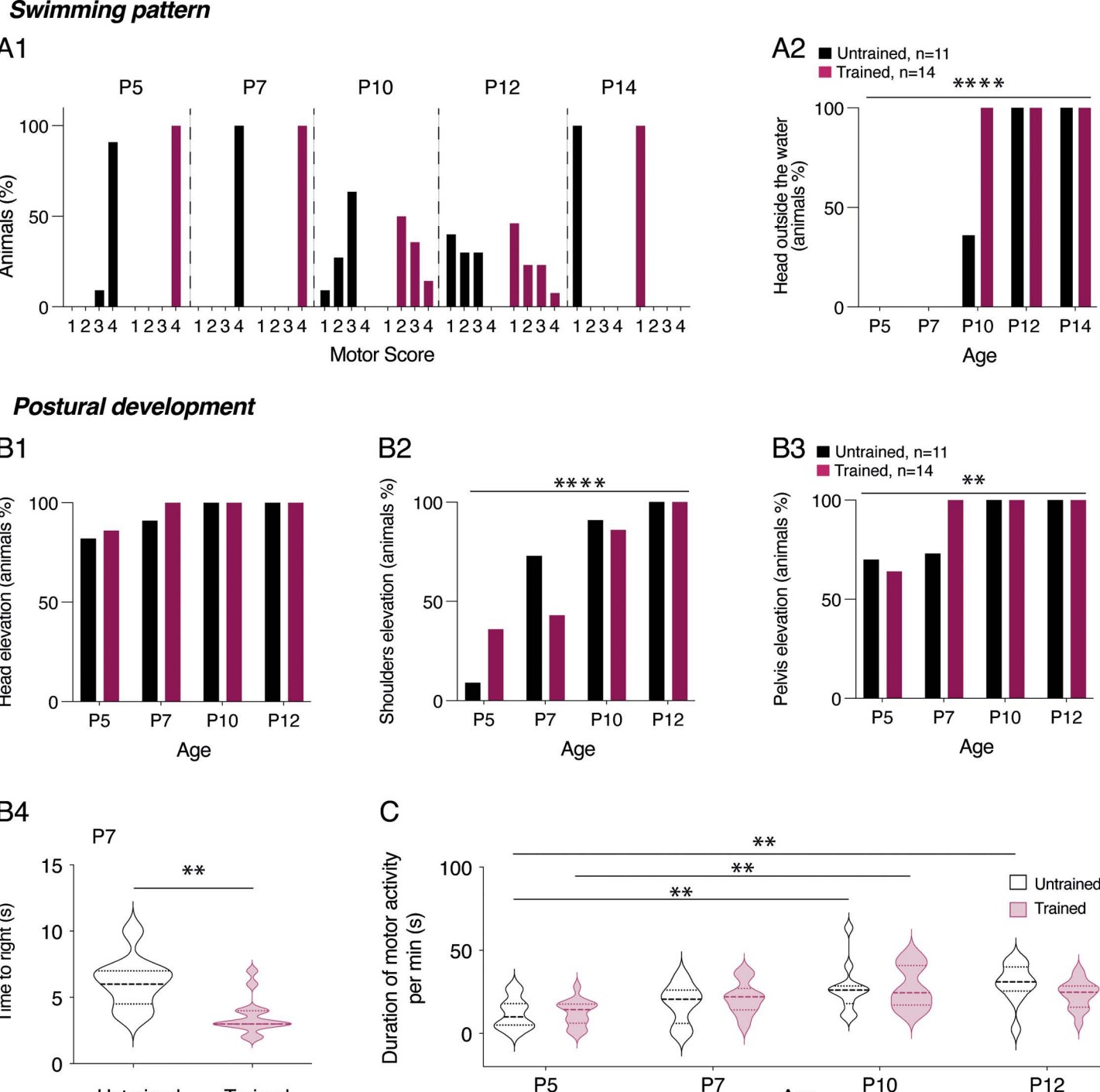

**Fig 11. Motor and postural development in trained pups during the first two postnatal weeks. A.** Histogram showing the percentage of untrained (black filled bars) and trained (purple filled bars) animals based on the motor score (Table 1 P5-P12) assessed during swimming between postnatal day 5 (P5) and P14 (A1). Percentage of untrained (black filled bars) and trained (purple filled bars) mice able to raise their heads above water during swimming as a function of age (A2). **B.** Percentage of untrained (black filled bars) and trained (purple filled bars) mouse pups able to raise their heads (B1), shoulders (B2), and pelvis (B3) on a solid surface as a function of age. **p < 0.05 and ****p < 0.0001, Chi-square test. Violin plots showing the time to right during the righting test in untrained (black, n = 11) and trained (purple, n = 14) P7 mice. **p < 0.01, Mann-Whitney test (B4). **C.** Violin plots showing the duration of motor activity per minute in untrained (black, n = 11) and trained (purple, n = 14) mice as a function of age. **p < 0.01, two-way ANOVA analysis, followed by uncorrected Fisher's LSD post-tests. Underlying data can be found in the S1 Data Sheet.

one-minute period, increased with age but was no significantly different between trained and untrained animals during the first two postnatal weeks (refer to Fig 11C).

These data indicate that the postural development of trained pups differs slightly but significantly from that of untrained ones during the first postnatal week and that these differences disappear during the second postnatal week.

## Discussion

In adult rodents, numerous studies have revealed that dynamic reconfigurations occur in the spinal networks in charge of the locomotor command when the motor activity is increased or modified [6,8–10,12,15–18,20,27,59–62]. The present study provides the first investigation of the impact of an increase in activity in the developing motor spinal cord and hindlimb muscles of newborn mice. The brief (2 sessions of 5 min per day during two days) swim training regimen we implemented, accelerated the acquisition of the four-limb swim pattern in trained P3 mice, thereby validating our training protocol as an effective method for inducing functional changes in the neuromuscular system. Our RNA-seq analysis, performed on laser-microdissected LMC, a region enriched in MNs, revealed changes in transcripts coding for transcription factors and homeobox genes after training. Similar effects were observed in the lumbar spinal cords of adult rodents engaged in voluntary wheel-running exercise [63,64], suggesting that expression of certain transcription factors is activity-dependent and specific to motor activity, regardless of the life stage. Among the differentially expressed genes after training, we observed numerous transcripts coding for transcription factors involved in inhibitory neuron development and differentiation, as well as for synaptic and scaffolding proteins. Anatomical and biochemical analyses will be required to confirm changes in protein expression, but we hypothesize that these changes underlie at least part of the differences we observed in the spinal network excitability after training.

In adults, different types of training have been shown to induce distinct transcriptional changes, with regulated genes involved in spinal cord development, neuronal positioning, signaling pathways, synaptic reorganization, pain transmission, immune response and cytoskeletal dynamics [63–65]. While our results highlight some similarly affected pathways, they also reveal that many differentially regulated genes are related to developmental changes in the neuromuscular system as a result of the motor training performed during the postnatal period.

Based on our RNAseq and behavioral data, we could hypothesize that swim training during early postnatal development accelerates the maturation of the neuromuscular system. However, as detailed below, when compared to the well-described development of MNs and muscles in rodents, the changes we report following motor training do not correspond to a simple general accelerated maturation.

During postnatal maturation, MN input resistance decreases [47] and ADP expression increases [36] which aligns with what we observed after training. However, it has been shown that rheobase increases and both AHP rise time and half decay time values decrease with MN development [10,36,47]. We observed no changes in the rheobase value after training, but we found an increase in total AHP duration and amplitude in trained MNs, with a slower repolarization phase duration (half decay time) without changes in the hyperpolarization phase (rise time). In adult rodents, the electrophysiological adaptations of MNs following different motor trainings have been described in details, revealing specific functional adaptations depending on the type of training tested [15,16,18,59,66]. Interestingly, in our experimental conditions, the AHP parameters of trained MNs evolved similarly to adult slow-MNs after long term wheel training [15], indicating that AHP, the main regulator of MN firing frequency is highly sensitive to moderate physical activity protocols. Although MN excitability typically decreases with age, and MNs switch from type I and II discharge profiles to type III and IV around P8-P9 [36,47,67], our training protocol did not alter overall excitability or firing profiles proportions. The reduction in AHP amplitude we observed, likely reduces the effort required to maintain repetitive firing in MNs potentially optimizing motor output and its efficiency as well as improving fatigue resistance [20].

We observed a change in gene expression of potassium channels, associated with an enhanced inwardly rectifying Kir current in trained MNs. It has been shown that various Kir subunit associations exhibit unique single channel conductance

[48,68,69]. Changes in homo or heteromeric assemblies of Kir 2.x subunits could therefore have occurred in trained MN, sustaining the higher inward Kir current we observed, although no changes in Kir 2.2 protein expression were detected. To the best of our knowledge, no data are available concerning Kir-current in developing lumbar MNs. As described in supraspinal structures [70], the amplitude of this current, which is a key determinant of neuronal excitability, should certainly increase during postnatal development in MNs.

One most striking effect of the training regimen used in the present study was its impact on ADSP expression at VLF-MN synapses. We previously showed that VLF-MN connections exhibit different ADSP profiles after HFS application to VLF fibers in spinal slices and that ADSP is differentially regulated between the first and second postnatal weeks in mice [34]. Specifically, VLF-HFS triggered LTD and STD at VLF-MN synapses in P1-P4 mice, while it elicited LTD, STD, or no plasticity (NP) at P8-P12. The transition from the two forms to the three forms of ADSP profile seems to occur just after P4 [34]. This is why the training protocol was performed over two days to enable the characterization of ADSP at P3 just before the transition occurs. ADSP expression at VLF-MN synapses depends on whether the MNs are associated with flexor or extensor muscles. STD was recorded in MNs connected to the ankle flexor *tibialis* muscle, while LTD and NP were observed in MNs innervating the ankle extensor *gastrocnemius* muscle [34]. In this study, we observed the emergence of the NP profile at VLF-MN synapses in trained P3 mice. At birth, extensor MNs are more immature than flexor MNs [71]. We hypothesize that the plasticity expressed at VLF-extensor MNs was changed to adapt to the increase in activity mediated by training. The physiological role of ADSP in limb MNs is currently unknown. The modulation of ADSP that we observed in a previous study [72] as well as its adaptation following motor training suggests that these plastic processes play a crucial role in the physiology of MNs.

The effects of the training regimen on myelin and axons we observed are in complete agreement with the previously demonstrated dependency of the myelination process on neuronal activity in the developing central nervous system [56]. Rather than a general increase in myelin sheath and axon area in the spinal cord, we demonstrated an area-specific impact of the training. Changes occurred only in motor-related areas, with an increase in both the myelin sheath and axon area in the ventrolateral column and, to a lesser extent, in the ventral commissure. Both increased diameter and enhanced myelination of the axons involved in motor function could lead to a more efficient transmission of electrical signals in motor spinal networks [73–75].

Early postnatal AChR clusters initially form oval-like plaques. Within the first three postnatal weeks, these plaques transform into a multiperforated, branch-like structure with a pretzel-like shape [76]. Additionally, NMJs exhibit a remarkable degree of plasticity in response to activity levels, with studies showing that different types of motor training can lead to various postsynaptic reconfigurations [77]. In our study, we observed significant changes in the morphology of AChR clusters between extensor and flexor muscles. As previously mentioned for ADSP, the greater immaturity of extensor hindlimb muscles compared to flexor muscles at birth [71] could explain this increased sensitivity to activity-induced changes. We found that the extensor posterior muscles showed an increase in both compactness and sphericity coefficients, suggesting a greater level of roundness in the AChR clusters, which is consistent with the observed decrease in the area-to-volume ratio. The area-to-volume ratio describes surface exchange efficiency in biology. Objects with a low ratio have a small surface area compared to their volume, typical of larger or more spherical objects. A lower area-to-volume ratio suggests more densely packed AChR clusters at NMJs, potentially enhancing neurotransmitter binding and synaptic signaling [78–81]. However, these compact and spherical AChR clusters differ from their normal development into complex pretzel-like structures, possibly reflecting NMJ adaptations to increased motor activity.

It should be underlined that the training we used in this study led to an atrophy of hindlimb muscles. Previous studies have shown that overtraining and insufficient recovery time can induce atrophy in rodents skeletal muscle fibers by increasing the protein catabolism/anabolism ratio [82,83]. Muscle adaptations to exercise typically involve changes in the contractile properties of motor units [84,85]. We did not observe changes in MyHC content. Training induces metabolic alterations in muscles, such as improved oxygen uptake, increased capillary density, larger and more numerous

mitochondria, associated with increased oxidative enzyme activity. We did not investigate muscle metabolism or strength, making it difficult to determine if hindlimb muscles exhibit improved contractile function and efficiency despite atrophy after training.

The impact of the short motor training regimen we imposed to pups could still be observed in the second postnatal week as evidenced by the behavioral and HPLC data. Specifically, the temporal sequence of acquiring the adult-like swimming pattern and the postural development of mouse pups differed between trained and untrained animals, with differences persisting until almost the second postnatal week. Moreover, the spinal contents of 5-HT and NA were found increased 8 days after the cessation of training in P10 trained pups. The chromatographic analysis we performed could not determine whether the density of descending fibers reaching the lumbar region increased after training in P10 spinal cords or if the release properties of monoaminergic fibers changed. Physical exercise has been shown to promote axonal regrowth from descending pathways after spinal cord injury in correlation with improved locomotor abilities [86]. Training might certainly promote descending monoaminergic pathway influences during motor spinal development by stimulating axon growth. Our transcriptomic data identified differentially regulated gene encoding the alpha-1D adrenergic receptor. Given their essential role in the development of spinal motor networks, further investigations are required to understand the histological and functional adaptations that monoaminergic systems exhibit following motor training.

In adults, training protocols of long duration are required to observe changes in spinal motor networks [15,16,18,20,60,66]. Our data show that a brief training protocol in early postnatal life can trigger significant functional adaptations in the neuromuscular system of developing mouse pups, which differ from a simple acceleration of maturation processes. Our study highlights that even mild changes in neural network functioning during critical periods of development can have a major impact, visible at both the macro (behavioral) and micro level (cellular and genomic). Therefore, careful studies should be conducted before using a motor activity to improve motor skill or locomotion acquisition, or as a tool for functional rehabilitation in the developing spinal cord and muscles. Increasing activity in the motor network should not be considered harmless.

## Materials and methods

### Animals and ethics

Experiments were conducted using 538 C57Bl/6JRJ from Janvier (Le Genest-Saint-Isle, France) and 10 B6.Cg-Tg(Hlxb9-GFP)1Tmj/J (HB9-GFP) mice from the Jackson laboratory (Bar Harbor, ME, USA), aged postnatal day 1 (P1) to P12, without sex discrimination. This study adhered strictly to the guidelines outlined in the European Committee Council Directive and the regulations set forth by the French Agriculture and Forestry Ministry for the care and use of animals in research. All experimental procedures were approved by the local ethics committee of the University of Bordeaux (approval number 2016012716035720) and French Ministry of Higher Education and Research. Every possible effort was made to minimize animal suffering and reduce the total number of animals utilized in this study.

### Training procedure

Newborn mice underwent swimming training twice a day, at 9 AM and 5 PM on postnatal days 1 and 2 (Fig 1A). Each pup was individually placed in the center of a Plexiglass tank (90 x 90 x 30 cm) filled with warm water at 37°C. The training protocol consisted of five spontaneous swimming activity sessions (S1 to S5), each lasting 15 seconds, separated by 45-second breaks (Fig 1A). If any signs of distress were observed during a session (such as cessation of swimming or drooping of the head), the mouse was promptly removed from water. After each session, mouse pups were carefully dried and warmed. Untrained mice were tested only once at one time point corresponding to one of the training time slots (P1 9 AM or P1 5 pm or P2 9 AM or P2 5 pm), with each test consisting of two 7-second sessions separated by a 45-second interval (Fig 1A). To evaluate swimming ability, we visually scored the number and types of limbs used during the first 7

seconds of each session (see Table 1 P1-P3). This training protocol is based on empirical observations of pup behavior in water. We observed that swimming sessions of 15 seconds marked the threshold at which P1 pups began to exhibit signs of fatigue and that a first session longer than 7 seconds triggers adaptations in swimming patterns during the second one.

## Motor and postural development analysis

One-minute videos were recorded for each pup moving freely on a clean and enclosed flat surface (90 x 60 cm) at P5, P7, P10, and P12 between 9 and 11 AM. To prevent rapid loss of body heat and separation issues, mice were removed from the dam for no more than 10 min. A piece of cotton with the nest odor was used to induce locomotor behavior in pups [87]. Head lifting, shoulder, and pelvis elevations were visually detected. The overall motor activity of each animal was assessed by timing all motor events (mainly pivoting, crawling, steps, early walking) during recording. P5-P12 pups were then placed in a warm water tank for 20 seconds to assess their swimming pattern and scored (see Table 1 P5-P12). The righting reflex of both trained and untrained mice was evaluated in P3 pups as well as in P5 to P12 mice. For this purpose, each animal was placed on its back and the time taken to right itself by completing a 180° turn was measured. The trial was considered "abandoned" if the animal failed to turn, ceased moving, and did not exceed 90 seconds.

## Intracellular recordings of motoneurons in spinal cord slices

P3 animals were anesthetized with 4% isoflurane until reflexes were lost and decapitated. The spinal cord was isolated in an ice-cold sucrose-based saline solution containing the following (in mM): KCl 2, $CaCl_2$ 0.5, $MgCl_2$ 7, $NaH_2PO_4$ 1.15, $NaHCO_3$ 26, glucose 11 and sucrose 205, equilibrated with 95% $O_2$ and 5% $CO_2$. The lumbar region was then sliced into 350 μm thick transverse sections. Slices were allowed to recover for at least 1 h at 30°C in oxygenated (95% $O_2$/5% $CO_2$) artificial cerebrospinal fluid (aCSF) composed of (in mM): NaCl 130; KCl 3; $CaCl_2$ 2.50; $MgSO_4$ 1.3; $NaH_2PO_4$ 0.58; $NaHCO_3$ 25; glucose 10. Whole-cell patch-clamp recordings from putative lumbar MNs identified by their relatively large soma in lamina IX, were performed using glass microelectrodes (3–6 MΩ) filled with (in mM): K Gluconate 120, KCl 20, $MgCl_2$ 1.3, EGTA 1, HEPES 10, $CaCl_2$ 0.1, GTP 0.03, AMPc 0.1, Leupeptine 0.01, $Na_2$-ATP 3, and 470.4 mg of D-Mannitol, at a pH of 7.3. All experiments were conducted at room temperature. Recordings were acquired using a Multiclamp 700B amplifier (Axon Instruments, CA, USA), with data being filtered at 10kHz and digitized through an interface (Interface ITC-18, Instrutech, Longmont, USA). Data acquisition and subsequent analysis were performed using Axograph X software.

Input membrane resistance (Rin) was determined by analyzing voltage-current relationships at −60 mV. After-hyperpolarization (AHP) parameters were computed after eliciting a single action potential (AP), with a brief 7 ms, 2.5 nA depolarizing current pulse. Firing behavior was studied using 500 ms square-wave depolarizing current pulses with increasing amplitudes. The instantaneous firing frequency ($f$-I) relationship was calculated using linear fitting of the first three data points. The different types of $f$-I patterns expressed in MNs during progressive triangular steps of both ascending and descending current ramps (range: 0.04–0.88 nA.s$^{-1}$) were classified as previously described [46]. Kir current was examined during long voltage ramps (−40 mV to −150 mV, 1 s duration). The potassium reversal potential (Erev) was determined at the knee-point of the Kir current curve for each cell. Inward and outward currents were measured at membrane potentials equidistant from Erev (+40 mV and −40 mV, respectively), and chord conductances (Gm) were calculated using the formula: Gm=I/ (Vm - Erev) [88]. Hyperpolarization-activated ($I_H$) current was elicited by a series of hyperpolarized voltage pulses (1 s duration) from −100 to −65 mV with 5 mV increments.

Synaptic responses and activity-dependent synaptic plasticity (ADSP) were evoked following established protocols [34]. Throughout recordings, polysynaptic transmission was minimized using a high-cation aCSF containing 7.5 mM $CaCl_2$ and 8 mM $MgSO_4$. GABAergic and glycinergic inputs were blocked by 1 μM of gabazine and 1 μM of strychnine. A bipolar tungsten electrode was positioned in the ventrolateral quadrant of the spinal cord slice to stimulate axons in the ventrolateral funiculus (VLF). Stimulation intensities ranged from 10 to 60 μA. Excitatory postsynaptic currents (EPSCs) were recorded from MNs held at −60 mV in voltage-clamp mode. After establishing a stable period of 10 min with VLF stimulations at 0.03 Hz, a 50 Hz high-frequency stimulation (HFS) was applied to VLF axons for a duration of 2 seconds (referred

to as VLF-HFS). During VLF-HFS, the intensity of VLF stimulation was increased to twice the baseline level and MNs held in current-clamp mode to facilitate normal depolarization and firing. VLF-EPSC amplitudes were normalized to the mean VLF-EPSC amplitude during the pre-HFS control period.

### Spontaneous synaptic event recordings

Spontaneous excitatory post-synaptic currents (sEPSCs) were recorded from MN held at −60mV in the voltage clamp mode in the presence of blockers for the glycinergic and GABAergic receptors, strychnine and gabazine (1 µM each), respectively. Spontaneous inhibitory post-synaptic currents (sIPSCs) from MNs held at −90 mV in the presence of the ionotropic glutamatergic blockers, DNQX and AP5 (10 µM each). After a 10 min period of stabilization, spontaneous synaptic currents were recorded over a 10 min period. The analysis of the frequency and amplitudes of spontaneous currents was conducted by examining 100 randomly selected events in each recorded MN.

### Laser capture microdissection and RNAseq

Spinal cords from untrained and trained P3 HB9-GFP C57BL/6JRJ mice were dissected from anesthetized animals, cryosectioned into longitudinal sections (20 µm), and collected on RNase-free microscope slides made of polyethylene-naphthalene membrane. Immediately after dehydration, GFP-positive MN somas were identified, and the lateral motor column (LMC) was dissected using a PALM laser microdissection and capture system (P.A.L.M. Microlaser Technologies AG, Bernried, Germany). The collection of LMC samples was limited to a maximum of 30 min per slide to minimize the risk of RNA degradation. The microdissected material was then treated with lysis buffer and stored at −80°C until RNA extraction using the *ReliaPrep RNA Cell Miniprep System* from Promega (Promega, La Farlede, France). RNA integrity and quantity were assessed using a Bioanalyser 2100 (Agilent Technologies, Massy, France) and a Nanodrop 1000 (Thermo Scientific, Waltham, USA), respectively. Only RNA samples with an RNA integrity number (RIN) greater than 7 were further processed. For RNA-Seq, the polyadenylated fraction of RNA was sequenced using the HiSeq 4000 system (Ilumina, San Diego, USA). Three biological replicates were used for both trained and untrained mice (n = 5 animals per group). Library preparation was done with 200 pg of cDNA employing the Nextera XT kit (Illumina, San Diego, CA, USA). Library quality and molarity were determined with the Qubit and Tapestation, using a DNA High sensitivity chip (Agilent Technologies, Santa Clara, CA, USA). Libraries were pooled and loaded for clustering on a single-read Illumina Flow cell (Illumina, San Diego, CA, USA), providing an average of 50 million reads per library. Reads of 100 bases were generated using the TruSeq SBS chemistry on an Illumina HiSeq 4,000 sequencer (Illumina, San Diego, CA, USA). To assess sequencing quality, FastQC v.0.11.5 was employed. The reads were mapped to the UCSC *Mus musculus* mm10 reference with the STAR aligner v.2.6.0c. Transcriptome metrics were evaluated with Picards tools v.1.14, and the count data were prepared using HTSeq v0.9.1. Differential expression analysis was conducted with the statistical analysis R/Bioconductor package edgeR v.3.18.1. Counts were normalized according to the library size and filtered. Genes with a count of at least 1 count per million reads (cpm) in a minimum of 3 samples were retained for the analysis, while genes with low or no expression were filtered out. The tests for differentially expressed genes were performed with a general linearized model (GLM) using a negative binomial distribution. The p-value for differentially expressed genes (< 0.05) were corrected for multiple testing errors with a 5% False discovery Rate (FDR), and a criterion of a 2-fold or greater difference was applied. To conduct Gene Ontology (GO) enrichment analysis on the generated datasets of differentially expressed genes, MetaCore (Clarivate Analytic, Philadelphia, USA) was employed.

### Western immunoblotting

P3 animals were anesthetized with 4% isoflurane until reflexes were lost and decapitated. Spinal cords were dissected and the ventral lumbar part of the spinal cord was isolated and stored at −80°C until use. The samples were then processed as previously described [89]. After protein quantification and transfer, the membranes were blocked in Tris buffer

saline 1X (TBS: Tris 10 mM, NaCl 200 mM, Tween-20,0.05%) supplemented with 5% milk for 60 min, then incubated overnight with the following antibodies: rabbit anti-Kir 2.2 (1:500, Table 5) and mouse anti-GAPDH (1:2000, Table 5) at 4°C. The membranes were then thoroughly washed and incubated with fluorescent secondary antibodies (in TBS 1X: goat anti-rabbit IRDye 800CW, goat anti-mouse IRDye 680RD, Li-Cor, Lincoln, USA). Specific GAPDH (37 kDa) and Kir 2.2 bands (48 kDa) were visualized with a Li-Cor Odyssey fluorescent detection system (Li-Cor, Lincoln, USA), and quantification was performed using Image studio software. For each sample, the intensity of the Kir 2.2 band was scanned and normalized to GAPDH expression.

## Immunohistochemistry

P3 mice were deeply anesthetized with ketamine/xylazine and transcardially perfused with a 0.1 M phosphate-buffer saline (PBS) solution followed by a 4% paraformaldehyde (PAF-PBS) solution. The lumbar part of the spinal cord was removed and postfixed in a 4% PAF-PBS solution overnight. The samples were cryoprotected in a 20% sucrose-PBS solution for 24 hours. For muscle tissue, P3 mice were anesthetized with 4% inhaled isoflurane for 5 min, decapitated and entire section of the leg between the knee and the ankle of the animal immediately removed. Both spinal cord and muscle samples were embedded in Tissue Tek O.C.T. Compound, flash-frozen in isopentane at −45°C, and stored at −80°C.

**Myelin sheath staining.** Coronal sections of lumbar spinal cord samples (14 μm thick) were washed in PBS for 3 x 10 min and then incubated with FluoroMyelin (Table 5) in a slide moisture chamber at room temperature (RT) for 15 min.

**Characterization of muscle fiber type.** Transverse sections of muscle samples (14 μm thick) were washed in PBS and transferred in a 1% BSA and 0.3% TritonX-100 solution in PBS for 90 min at RT. To quantify myofiber profiles and delineate myofiber membrane, anti-myosin and anti-laminin primary antibodies, respectively (Table 5) were incubated overnight in a humidified chamber at RT. After PBS washes, slices were incubated for 90 min with the following secondary antibodies: Alexa-568 donkey anti-mouse and Alexa-488 donkey anti-rabbit (1:500, Thermo Fisher, France)

**Neuromuscular junction (NMJ) staining.** Longitudinal sections (30 μm thick) of the entire lower hindlimb of pups were rehydrated in PBS for 2 min and post-fixed in a 2% PAF-PBS solution for 10 min. After a quick PFA removal rinse, the sections were washed and treated with a 0.1M glycine solution in PBS to block aldehyde residues for 30 min at RT with gentle agitation. Tissues were then incubated with a 2% BSA and 0.3% TritonX-100 in PBS solution for 90 min at RT. The sections were then incubated overnight in a slide moisture chamber at RT with the primary antibodies: mouse anti-neurofilament and rabbit anti-laminin (Table 5) diluted in the blocking and permeabilization solution. After PBS washes, sections were incubated in a dark, humidified chamber at RT for 90 min with the following secondary antibodies: Alexa-568

**Table 5. Primary antibodies, molecular probes and conjugates used for IHC and WB experiments. MHC: Myosin Heavy Chain. AChR: Acetylcholine Receptor. GAPDH: glyceraldehyde-3-phosphate dehydrogenase.**

| Target | Antibody/ Probe | Host Animal | Dilution | Distributor, Cat. Num. |
|---|---|---|---|---|
| Myelin | FluoroMyelin Green | – | 1:300 | Invitrogen; (F34651) |
| MHC, Embryonic | BF-G6 (supernatant) | Mouse | 1:100 | DHSB, Iowa, USA |
| MHC, Type I | BA-D5 (supernatant) | Mouse | 1:100 | DHSB, Iowa, USA |
| MHC, Type IIA | SC-71 (supernatant) | Mouse | 1:100 | DHSB, Iowa, USA |
| MHC, Type IIB | BF-F3 (supernatant) | Mouse | 1:100 | DHSB, Iowa, USA |
| Membrane | Anti-Laminin | Rabbit | 1:1000 | Sigma-Aldrich; (L9393) |
| Neurofilament | 3A10 (supernatant) | Mouse | 1:500 | Invitrogen; (B13422) |
| Nicotinic AChR | α-Bungarotoxin Alexa 488 | – | 1:700 | DHSB, Iowa, USA |
| KIR2.2 receptor | Anti-K$_{ir}$ 2.2 (KCNJ12) | Rabbit | 1:500 | Alomone (APC-042) |
| GAPDH | Anti-GAPDH | Mouse | 1:2000 | Abcam (AB8245) |

donkey anti-mouse, Alexa-647 donkey anti-rabbit (Thermo Fisher, France), and the α-Bungarotoxin Alexa 488 conjugate (Table 5).

Regardless of the labelling performed, slides were mounted with coverslips using Vectashield Hard Set medium (Euro-bio, France).

## Microscopy and data analysis

**Myelin analysis.** All images were acquired using an epifluorescence microscope (Olympus, Japan; 1344 x 1024 pixels) with a 20x air objective and z stacks intervals of 1 μm. Each myelin sheath was manually selected using the Fiji ROI manager. Subsequently, the Fiji "Ridge Detection" plugin was applied with the following parameters: line width = 20; high contrast = 230; low contrast = 100. Only completely detected rings of myelin sheath were included in the analysis. We measured the total myelinated axon area that includes both the axon and myelin sheath, the myelin sheath area and width. The axon area was calculated by subtracting the myelin sheath area from the total myelinated axon area.

**Muscle fiber detection.** All images were acquired using a confocal microscope (LSM 900, Zeiss, France) with a 63x oil immersion objective and z stacks intervals of 0.22 μm with the Zeiss Zen software. A polygon selection was used to choose the muscle to analyze. Subsequently, several Fiji processes were applied on the laminin channel, including the "Tubeness" function (σ = 0.3244), "Enhanced contrast" processing (0.1% saturated), "Gaussian Blur" filter with σ = 6, "Subtract background" (rolling = 100), and finally, "Find Maxima" with parameters: noise = 20 and output type set to segmented particles with a light background. Using this method, individual myofibers were isolated and measurements of myofiber area, circularity, and solidity values were generated using Fiji. Circularity and solidity values were exploited to filter out incorrectly detected myofibers. The Fiji plugin "circularity" calculates an object's circularity using the formula: $circ = 4pi \, (area/perimeter^2)$, where a value of 1.0 indicates a perfect circle, and values approaching 0.0 suggest an increasingly elongated shape. The Fiji plugin "solidity" evaluates how smooth and convex the object's contour is, using the formula: $solidity = area/convex \, area$. A perfectly convex object will have a value of 1. In this context, detected particles with circularity and solidity values lower than 0.62 and 0.85, respectively, were excluded. This method resulted in the exclusion of approximately 23.5% ± 0.3 (n = 60 muscle samples) of detected myofibers in the muscles of untrained mice and 24.0% ± 0.4 (n = 50 muscle samples) of detected myofibers in the muscles of trained mice, with no significance differences observed (Mann-Whitney test, p = 0.47). For each accurately detected myofiber, Fiji generated a pixel mean measurement of the myofiber's interior in the myosin staining channel. Subsequently, a threshold for the minimum pixel mean value was established to determine whether a myofiber was considered positively labeled. This method was applied separately for each myosin type and the percentage of positively labeled myofibers was calculated relative to the total number of accurately detected myofibers in each muscle.

**NMJ analysis.** All images were acquired using a confocal microscope (LSM 900, Zeiss, France) with a 63x oil immersion objective and z stacks intervals of 0.2 μm. The morphology of AChR plaques was assessed using the Fiji software.

## Chromatographic analysis

Tissue levels of noradrenaline (NA), dopamine (DA) and serotonin (5-HT) were determined using an HPLC-ECD system as previously described [90]. The mobile phase composition was as follows (in mM): 60 $NaH_2PO_4$, 0.1 Disodium EDTA, 2-Octane-sulfonic acid in deionized water (18 MΩ/cm²) containing 7% methanol. To ensure accurate separation of the eluents in the chromatogram, the pH was adjusted to approximately 4 with orthophosphoric acid. Detection was carried out using a coulometric cell (5011 coulometric cell, ESA, Paris, France) coupled to a programmable detector (Coulochem II, ESA). The potentials of the electrodes were set at − 270 and + 350mV. Electrochemical signals were recorded *via* an interface (Ulyss) on a computer equipped with the Azur software (Toulouse, France). Calibration curves were generated

using standard solutions containing known concentrations of NA, DA and 5-HT, which were systematically injected each day prior to the analysis of sample series.

## Statistical analysis

Statistical analyses were conducted on raw data using GraphPad Prism software (Graphpad, CA, USA). Mann-Whitney or T-test, depending on the normal distribution of the data, were used to compare two datasets. Chi-square was used for contingency analysis, Kolmogorov-Smirnov for spontaneous synaptic events, and two-way ANOVA followed by uncorrected Fisher's LSD for further comparisons. All data are expressed as means ± SEM. Statistical significance level was set at $p < 0.05$. In the various figures, violin plots display the distribution of data along with the median and quartiles.

## Supporting information

**S1 Fig. A.** Top 10 GO enrichment network processes and **B**. Top 10 of GO processes, associated with upregulated genes, downregulated genes, or all the dysregulated genes in trained mice as ranked by the p-value (dashed line). The size of the circles reflects the number of genes involved in the processes described, while the color scale indicates the FDR value.
(TIF)

**S1 Video.** Representative movie of the swimming pattern of a P3 trained mouse pup (on the left) and a P3 untrained mouse pup (on the right).
(MP4)

**S1 Data. Data sheet: replication data file.**
(XLSX)

**S1 Table. Differentially expressed genes between trained- and untrained lateral motor column of P3 mice.** Over expressed genes are in blue. Under expressed genes are in Red.
(DOCX)

**S1 File. Raw image: Raw Kir2.2 (green) and GAPDH (red) bands from ventral lumbar spinal cords of 5 untrained (first 5 from the left) and 5 trained mice obtained by western blot.**
(TIF)

## Acknowledgments

The authors warmly thank A. Fayoux and G. Courtand for their technical help. Transcriptomic experiments were done at iGE3Genomics Platform, University of Geneva Switzerland. This work benefited from the support of the Laser Microdissection capture facility thanks to M. Maitre and H. Doat of the NeuroCentre Magendie Inserm U1215.

## Author contributions

**Conceptualization:** Sandrine S Bertrand.

**Data curation:** Camille Quilgars.

**Formal analysis:** Camille Quilgars, Florence E Perrin, Sandrine S Bertrand.

**Funding acquisition:** Sandrine S Bertrand.

**Investigation:** Camille Quilgars, Eric Boué-Grabot, Jean-René Cazalets, Florence E Perrin, Sandrine S Bertrand.

**Methodology:** Camille Quilgars, Eric Boué-Grabot, Philippe De Deurwaerdère, Florence E Perrin.

**Project administration:** Sandrine S Bertrand.

**Supervision:** Eric Boué-Grabot, Philippe De Deurwaerdère, Florence E Perrin, Sandrine S Bertrand.

**Validation:** Sandrine S Bertrand.

**Writing – original draft:** Camille Quilgars, Sandrine S Bertrand.

**Writing – review & editing:** Camille Quilgars, Eric Boué-Grabot, Philippe De Deurwaerdère, Jean-René Cazalets, Florence E Perrin, Sandrine S Bertrand.

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
