## [Editor Report · Decision Letter 0]

26 Nov 2024

Dear Dr Bertrand,

Thank you for submitting your manuscript entitled "When Little Means a Lot: Impact of a Brief Early-life Motor Training on Mouse Neuromuscular Development." for consideration as a Research Article by PLOS Biology.

Your manuscript has now been evaluated by the PLOS Biology editorial staff, and I am writing to let you know that we would like to send your submission out for external peer review.

Once your full submission is complete, your paper will undergo a series of checks in preparation for peer review. After your manuscript has passed the checks it will be sent out for review. To provide the metadata for your submission, please Login to Editorial Manager (https://www.editorialmanager.com/pbiology) within two working days, i.e. by Nov 28 2024 11:59PM.

Kind regards,

Taylor

Taylor Hart, PhD,

Associate Editor

PLOS Biology

thart@plos.org

---

## [Decision Letter · Decision Letter 1]

17 Jan 2025

Dear Dr Bertrand,

Thank you for your patience while your manuscript "When Little Means a Lot: Impact of a Brief Early-life Motor Training on Mouse Neuromuscular Development." was peer-reviewed at PLOS Biology. It has now been evaluated by the PLOS Biology editors, an Academic Editor with relevant expertise, and by several independent reviewers.

In light of the reviews, which you will find at the end of this email, we would like to invite you to revise the work to thoroughly address the reviewers' reports.

You’ll see that the reviewers think that the study is well executed and find the results of interest. All three reviewers had issues with the level of clarity in the methods, results, and data items, particularly the lack of clarity in how many animals were used in each experiment. R2 and R3 both found the analysis of gene expression changes after training to be of limited insight and suggested a more thorough analysis to justify the inclusion of this data. R3 finds issue with the logical flow throughout the manuscript, while still being quite positive of the paper overall.

We would like to invite a Major Revision of this manuscript before it can be accepted for publication. In particular, the revised manuscript should include additional analysis of the gene expression data as part of a new supplemental item in line with R2's and R3's suggestions, along with videos showing the swimming patterns as indicated by R3. In addition, you should carefully review and modify the data presentation and text to improve the clarity and correct any errors in line with the reviewer's comments.

Given the extent of revision needed, we cannot make a decision about publication until we have seen the revised manuscript and your response to the reviewers' comments. Your revised manuscript is likely to be sent for further evaluation by all or a subset of the reviewers.

**IMPORTANT - SUBMITTING YOUR REVISION**

*Re-submission Checklist*

*Published Peer Review*

*PLOS Data Policy*

*Blot and Gel Data Policy*

Sincerely,

Taylor

Taylor Hart, PhD,

Associate Editor

PLOS Biology

thart@plos.org

REVIEWS:

Reviewer #1: This study explored the influence of increased motor activity in newborn mice during early period of development of spinal cord circuitry and hindlimb muscles. The main conclusion is that a very short motor training performed just after birth is able to induce functional adaptation in the developing

neuromuscular system that could persist several days. The study is impressive in the scope and a number of electrophysiological, morphological and biochemical analyses performed, and the work is very well done. In vitro recordings on spinal cord slices are challenging, and the authors amassed a convincing number of recordings on which they base their conclusions.

The study is implemented with the high degree of rigor and technical excellence. Analyses are carefully performed and presentation is clear in general. The results are definitely interesting and a bit surprising, considering a very short-lasting training sessions. I have no significant criticisms. However, I would advice the Authors to reconsider presentation of all the detailed data in the tables and figures. A number of figures and plots is overwhelming, and some of them repeat the same data already presented in tables 2 and 3. It is a question whether to show all the insignificant data, which are shortly mentioned in the Results, but not discussed further.

The minor points which should be corrected or clarified:

- it is difficult to find a number of mice used in particular groups for analyses, a total number, division into trained and untrained groups; sometimes a number of mice is indicated in the text, but frequently it is not, and this makes more difficult to assess the importance of particular changes among a substantive number of data;

- a number of samples in Table 5 as well as a number of recordings used for violin plots might also be indicated in the plots or in the legends to make them more informative; presentation of a percentage in charts is informative, but should also be supported with a number of neurons/samples (Figs. 1BC, 3C2, 4D, 10C, 11);

- several data in Table 3 and 4 need to be corrected (rheobase is presented in nA? AP rise time in ms? current max in pA?);

- Fig. 3A1: is the scale bars (50ms and 2000pA) real for the presented record? It is stated in methods that "AHP parameters were computed after eliciting a single action potential (AP), with a brief 7 ms, 0.25 nA depolarizing current pulse". Basing o the scale bars the pulse here seems to be 2.5nA and duration at least twice longer, the same doubt concerns the AP and AHP duration. It is difficult for a reader to check such details in each presented chart, but when such a mistake has been found once, I would encourage the Authors to check such and similar details throughout the paper and all figures;

- I have concern about clear outliners visible in some of the violin plots showing distribution of data (especially in Fig. 3A5, Fig. 4 C2 and C3). Are the Authors sure that these measurements were based on reliable recordings, not suffering from an erratic observation or instability of a preparation? Could this be connected to slow-fast differences between MNs, and obviously, slow and fast MNs were not distinguished?

Reviewer #2: In this work, the authors designed a short motor training regimen at P1 and P2 to decode the impact of early training on the development of spinal motor neurons and hindlimb muscles using the combined techniques including behavioral analysis, RNA-seq, electrophysiological recordings, immunostaining, et al. They demonstrated that a short training caused many changes in newborn mice including swimming performances, gene expression, electrophysiological characteristics of spinal motor neurons, muscle development. The study is interesting, but the authors should strengthen the logical explanation of the changes they identified in different aspects.

I have some comments as below:

1. The author should give a brief explanation of the duration time designed in the training paradigm since that 7 sec or 15 sec were very short.

2. In Fig. 1B1�why are the numbers of animals in different test batches different within the same group (untrained or trained)? In the untrained group, the numbers were 64, 66, 61, and 61 at the trial 1, 2, 3, and 4 respectively. In the trained group, they were 110, 99, 103, and 91 at the training 1, 2, 3, and 4 respectively. Why did the authors use such a large number of animals for behavioral screening?

3. In Fig. 1B1 and Fig. 1B2: the changes were strange: At trial 1 of untrained mice, the proportion of mice unable to swim at session 2 increased significantly compared with that at session 1. Does it mean that P1 pups lose swimming ability after a 45-second break?

4. The authors studied training-induced gene expression changes using RNA-seq at P3. The molecules involved in the key signaling pathways should be deeply analyzed combining with the electrophysiological characteristics and morphological changes of training-induced neurons. For example, downregulated DEGs were clustered in neurogenesis, synaptogenesis, developmental NMJ, et al. Which DEGs could support the findings in changes of electrophysiological and morphological characteristics after training? Likewise, upregulated DEGs were clustered in many important biological processes, and these results should be further illustrated and confirmed.

5. At page 21 Line 12-13: "The proportion of MNs expressing LTD, STD, or No Plasticity differed significantly between untrained and trained mice…" What conclusion could be made from this finding? The authors should summarize at the end of this paragraph.

6. The authors described: a significant decrease in the amplitude of sEPSCs, less frequent but larger sIPSP events in trained MNs compared to the untrained MNs. In Fig. 6B2, the frequency of sIPSPs looked higher in the trained MNS (red dots) than the untrained MNs (black dots)? In Fig. 6B3, the amplitude of sIPSPs looked smaller in the trained MNS (red dots) than the untrained MNs (black dots)? Is there any morphological evidence supporting this finding?

7. The authors found a notable increase of 5-HT and NA between P3 and P10 in trained mice, I was wondering if it was possible to present any differences of their fiber density using immunofluorescent staining spinal sections.

Reviewer #3: Review Plos Biology : PBIOLOGY-D-24-03358R1

Title: When Little Means a Lot: Impact of a Brief Early-life Motor Training on Mouse Neuromuscular Development.

Authors : Quilgars et al

Overall comments

The authors compared the impact of an early swim training (5 sessions of 15 sec, twice a day for 2 consecutive days) in neonatal mice pups, compared to untrained pups (2 sessions of 7 sec, twice a day for 2 consecutive days for swimming assessement) on several structural and functional features in the spinal cord and in the interactions between motneurons and muscles. Briefly, they found changes in the acquisition of swimming abilities (acceleration in trained pups), no changes in the intrinsic properties of motoneurons from the lumbar spinal cord except a few aspects of membrane excitability, changes in some features of currents and synaptic plasticity, down- and upregulated genes involved in membrane properties of motoneurons and others structural aspects, changes in some neurotransmitters, and alterations of axonal myelination in the motor regions of the spinal cord. They also found changes in the morphology, phenotype after such an early and short motor training. All the different features studied here are very interesting and correspond to a huge stack of data and work, but sound like a juxtaposition, so they need to be more linked together into an integrated whole study.

The study is very sound and interesting, well-conceived, well-done and well-written but several mistakes and typos are present. The Results part on swim motor training lacks of precision and details that need to be clarified (see description below), otherwise the remaining parts of the manuscript are sound. The manuscript needs more work, precision and clarifications since it really deserves to be published in Plos Biology with a high impact in the field of developmental neuroplasticity. One can deplore the absence of working hypothesis in the Introduction that may link and justify the methods used to test it, and the use of too many abbreviations. English and typos should be corrected.

Abstract

From the Results and Fig 1A, the faster acquisition of swimming was assessed at P1 and P2 in trained and untrained pups; please clarify timing of results.

Methods

The description of the statistics used and their choice should be increased and justified. For example, I don't understand the use of Chi square in many comparisons and make some statistical results unclear, like the numbers of motoneurons expressing STD, LTD or no plasticity. Statistical test description needs to be added in the Results, like the value of Mann-Whitney tests and Spearman…

Results

Part 1: "Motor training accelerated the acquisition of the four-limb motor pattern for swimming."

The Methods, Results, Fig 1 from A to D and the corresponding legend are unclear and hard to follow and decipher, so they need to be clarified as this part is crucial for understanding the whole manuscript.

In Cazalets et al (1990), is written page 127: "During the first 24 hours, only 2/3 of the [rat] pups were swimming, whereas at day 2 they were all able to swim and move around the tank. … As all of them [rat pups] were not able to swim at P0, we did not perform quantitative analysis." The patterns of swimming of mice pups seems critical to be better defined in the present study, compared to rats that served as reference in previous reports (Cazalets et al, 1990) and from the "patterns H0 to H2" in mice pups in Cazalets (2000). Do the patterns of swimming differ between the various strains of mice used in the present study and from "C3H control strain" in Cazalets et al (2000). This is preponderant and crucial for the rest of the study. In addition, the authors use two strains of mice in the present study, the use of each one in the different experiments needs also to be clarified.

How many mice were used in the present study? Why don't the authors show also the numbers of mice for scores 2 to 4? Why the numbers in session 1 but not in session 2? All this part on the "motor training" and "swimming patterns" needs to be clarified from the Methods, Results and corresponding Fig 1 and legend. Why not to add videos of patterns in supplemental data? Supplemental videos may clarify the "non well-defined patterns of swimming" and each pattern of swimming

What is the interest of fig Fig 1C?

The numbers of mice used appears for the body weight comparison. It should appear before in the manuscript, as well as the repartition of the two strains, as previously suggested.

The maximal duration of the righting reflex session should also be described in Methods for ethical consideration since the authors wait for rats to be exhausted in this task (i.e. time to abandon in Fig1D).

The conclusion of this part of the Results may include the difference of training between untrained that performed 7 sec of swimming twice, 2 times a day for 2 consecutive days.

Actually, the part on top of Fig 1A for untrained mice (Trial 1 - Mouse 1 and so on...) is not clear and does not correspond to any description in the legend, Methods or Results. It suggests that each trial was made with different mice. This needs to be clarified.

The difference in swim training between the two groups needs to be clarified and emphasized along the manuscript. I don't understand why the authors used untrained control rats to assess swimming at P1 and P2 (corresponding to a "training") while they did it later from P5 to P14, which may have been enough and leaving untrained control pups really untrained?

Table 1 on Motor Scores. Why is there no score of 1, as it could be 0 for no swim and 1 for swim with 1 limb? Please justify and clarify.

The authors describe the exclusive use of hind limbs around P15, so why do the use of the hind and forelimbs are mixed in scoring from P5 to P12? Actually, the gradation of scores of this scale is not completely clear; please justify.

Part 2: "Motor training induces changes in gene expression in the lateral motor column."

There are discrepancies in the numbers of genes up -and downregulated between text and Fig 2.

The description of the changes in genes is too short and lacks of details to get the insights, only a summary of the results is produced. Maybe more details could be provided in Supplemental data? This raises the question of its interest as it has not been justified previously.

To me, there are also discrepancies between statistics in the text corresponding to the Fig 4C2 and C3. Please clarify.

Last part of Results: "Early motor training impacts the development of postural control during the two first postnatal weeks"

I don' agree with authors on the summary of results: "These data indicate that the postural development of trained pups differs slightly but significantly from that of untrained ones during the first postnatal week and that these differences disappear during the second postnatal week." To me, these results appear stronger than they described and should be emphasized more positively as an improvement and acceleration of the acquisition of sensorimotor abilities, as written in the Abstract.

Discussion

The different parts are very interesting and well-written and suited, but to me, several positive results are underestimated, like the "absence" of changes in the muscle phenotype and others. As previously suggested, all these very interesting results lack of integration and links together to serve a hypothesis on the impact of such an early and short motor training on so many structural, morphological and functional properties for the development and exercise-driven plasticity of the sensorimotor system.

These comments are meant to help the authors to improve the quality of the manuscript, which deserves for me to be published in Plos Biology

---

## [Decision Letter · Decision Letter 2]

17 Mar 2025

Dear Dr Bertrand,

Thank you for your patience while we considered your revised manuscript "When Little Means a Lot: Impact of a Brief Early-life Motor Training on Mouse Neuromuscular Development." for publication as a Research Article at PLOS Biology. This revised version of your manuscript has been evaluated by the PLOS Biology editors, the Academic Editor and the original reviewers.

Based on the reviews, we are likely to accept this manuscript for publication, provided you satisfactorily address the following data and other policy-related requests:

- TITLE: We would like to suggest a different title to increase accessibility for our broad audience: "Brief early-life motor training induces long-lasting changes in behavior and neuromuscular development in mice"

- FINANCIAL DISCLOSURE: Please add the links to the funding agencies in the Financial Disclosure statement in the manuscript details

- DATA POLICY:

Fig 1 B-D

Fig 2 all panels

Fig 3 A2-A6, B2-B3, C1-C2

Fig 4 A2-A3, B, C2

Fig 5 B-D

Fig 6 A2-A3, B2-B3

Fig 7 B-D

Fig 8 B1-B3

Fig 9 B1-B5, C

Fig 10 B1-B3, C1-C3

Fig 11 A1-A3, B1-B4, C

- DATA AVAILABILITY: For the sequencing data, you should deposit the raw data in a publicly available repository, and mention this in the data availability statement as well as providing a link or DOI.

- CODE POLICY

- ORIGINAL GELS: In line with our gel and blot reporting requirements, you should provide uncropped images of all Western blots or other gels.

We expect to receive your revised manuscript within two weeks.

*Published Peer Review History*

*Press*

Sincerely,

Taylor

Taylor Hart, PhD,

Associate Editor

thart@plos.org

PLOS Biology

Reviewer remarks:

Reviewer #1: I would like to thank the authors for addressing my points in the revised version of the manuscript. The current version is improved and all of my concerns have been considered.

Reviewer #2: The authors carefully addressed my concerns, and I have no further comments.

Reviewer #3 (J-Olivier Coq): Review Plos Biology : PBIOLOGY-D-24-03358R2

Title: When Little Means a Lot: Impact of a Brief Early-life Motor Training on Mouse

Neuromuscular Development.

Authors : Quilgars et al

General comments

The authors have performed most of the changes recommended and have considered and discussed the other changes suggested. The different changes requested by the three reviewers were complementary and the revised version R2 fits to me with the requirements and guidelines for publication in Plos Biology.

---

## [Editor Report · Decision Letter 3]

31 Mar 2025

Dear Dr Bertrand,

Thank you for your patience while we considered your revised manuscript "Impact of a Brief Early-life Motor Training on Mouse Neuromuscular Development." for publication as a Research Article at PLOS Biology. This revised version of your manuscript has been evaluated by the PLOS Biology editors, and we have noted a few more items to address before your paper can be formally accepted.

----

***Regarding the title, we understand your point and agree that it is best not to say that the changes are 'long-lasting'. The alternative that you have proposed, 'Brief early-life motor training induces changes in behavior and neuromuscular development in mice' is acceptable to us. But we also propose a slight modification 'Brief early-life motor training induces behavioral changes and alters neuromuscular development in mice'.

Whichever version you choose, please change the title in the manuscript file and the meta data to match when you submit the revision.

***Regarding the data, thank you very much for making your sequencing and other source data available. However, there are a few points on this topic still to address.

First, we found that the Research Data Gouv only guarantees maintaining the data for a minimum of 5 years (https://recherche.data.gouv.fr/en/faq). In the interest of maintaining access to your RNAseq data over the long term, we request that you also upload your sequencing data to the GEO database, as this is considered more standard (https://www.ncbi.nlm.nih.gov/geo/).

Second, I have examined the replication data for Figure 1 and quickly noticed what appear to be discrepancies between the plotted data and the corresponding part of the excel file.

- In Fig. 1B in the excel file, the data in the first column sum to 64, which matches the number shown in the figure above the first bar. However, the data in the second column sum to 110, whereas the number above the corresponding bar in the figure is 241. The first 'trained' bar in the figure shows about 36%, which would about match to the raw count of 38 in the file, but only if the total count is indeed 110 - if the count is 241, then there is a substantial mismatch. The same issue appears throughout the figure. It isn't clear to me whether the numbers listed above the histogram bars are wrong, or if the count data in the excel file are wrong, or if the data is displayed incorrectly in the chart.

- For Fig. 1E, there is data for this in the supplemental excel file but there is no panel in the figure with this label. Instead, the second part of Fig. 1D appears to match this data.

I have not comprehensively examined how well the replication data you provided match your figures, so there may be more discrepancies. Please address these issues and also double-check that all data are displayed correctly and that the figures and replication data files match throughout.

***In addition, the figure legends require a bit more information. Please specify in the figure legends where each piece of source data may be found among the different files you uploaded to the repository. It may help to adopt a consistent naming scheme for the source data files, for example "S1_Data.xlsx," which you can then reference in the figure legends parenthetically as ("S1 Data").

Also, some details of the figures need to be explained explicitly in the figure legends. For example, the legends should say what the numbers shown above some of the histogram bars in Fig. 1B represent. The dotted lines shown on the violin plots should also be explained in the legends. This will also make it easier to sort out any other discrepancies.

-----

We expect to receive your revised manuscript within two weeks.

*Published Peer Review History*

*Press*

Sincerely,

Taylor

Taylor Hart, PhD,

Associate Editor

thart@plos.org

PLOS Biology

---

## [Editor Report · Decision Letter 4]

4 Apr 2025

Dear Dr Bertrand,

Thank you for the submission of your revised Research Article "Brief early-life motor training induces behavioral changes and alters neuromuscular development in mice" for publication in PLOS Biology. On behalf of my colleagues and the Academic Editor, Jacques-Olivier Coq, I am pleased to say that we can in principle accept your manuscript for publication, provided you address any remaining formatting and reporting issues. These will be detailed in an email you should receive within 2-3 business days from our colleagues in the journal operations team; no action is required from you until then. Please note that we will not be able to formally accept your manuscript and schedule it for publication until you have completed any requested changes.

PRESS

Sincerely, 

Taylor Hart, PhD,

Associate Editor

PLOS Biology

thart@plos.org